# NEURAL STOCHASTIC DIFFERENTIAL EQUATIONS FOR UNCERTAINTY-AWARE, OFFLINE RL

**Cevahir Koprulu**[1]* **Franck Djeumou**[2]* **Ufuk Topcu**[1]
[1]The University of Texas at Austin [2]Rensselaer Polytechnic Institute

## ABSTRACT

Offline model-based reinforcement learning (RL) offers a principled approach to using a learned dynamics model as a simulator to optimize a control policy. Despite the near-optimal performance of existing approaches on benchmarks with high-quality datasets, most struggle on datasets with low state-action space coverage or suboptimal demonstrations. We develop a novel offline model-based RL approach that particularly shines in low-quality data regimes while maintaining competitive performance on high-quality datasets. *Neural Stochastic Differential Equations for UNcertainty-aware, Offline RL* (NUNO) learns a dynamics model as neural stochastic differential equations (SDE), where its drift term can leverage prior physics knowledge as inductive bias. In parallel, its diffusion term provides distance-aware estimates of model uncertainty by matching the dynamics' underlying stochasticity near the training data regime while providing high but bounded estimates beyond it. To address the so-called model exploitation problem in offline model-based RL, NUNO builds on existing studies by penalizing and adaptively truncating neural SDE's rollouts according to uncertainty estimates. Our theoretical results show that penalization via a distance-aware uncertainty estimator incentivizes the policy to stay close to the offline data. Our empirical results in D4RL and NeoRL MuJoCo benchmarks evidence that NUNO outperforms state-of-the-art methods in low-quality datasets by up to 93% while matching or surpassing their performance by up to 55% in some high-quality counterparts.

## 1 INTRODUCTION

Offline reinforcement learning (RL) concerns the problem of learning control policies from offline datasets of interactions (Lange et al., 2012; Levine et al., 2020). This paradigm captures safety-critical real-world settings such as healthcare (Tseng et al., 2017; Wang et al., 2018), robotics (Levine et al., 2018; Rafailov et al., 2021) and autonomous driving (Yu et al., 2020a), where logged data is abundant, simulators are computationally expensive, or online learning causes hazardous behavior. Although off-policy RL algorithms can, in principle, address settings with a priori available data, they fail in the offline setting due to the distribution shift between the dataset and learned policies (Fujimoto et al., 2019; Kumar et al., 2019). To resolve distribution shift, model-free offline RL methods introduce conservatism via constraining learned policies to available data (Jaques et al., 2019; Wu et al., 2019; Fujimoto & Gu, 2021) or penalizing out-of-distribution actions (Kumar et al., 2020; Bai et al., 2022). However, such approaches struggle with sub-optimal behavior policies (Yu et al., 2020b).

Offline model-based RL trains a control policy via synthetic data generated by a learned dynamics model (Kidambi et al., 2020; Yu et al., 2021). Compared to offline model-free RL, employing the learned model improves generalization beyond the training data regime Rigter et al. (2022). However, naive application of model-based RL causes a phenomenon called *model exploitation*: Learned control policies exploit the parts of the state-action space where the model is inaccurate and overestimates the return Janner et al. (2019); Yu et al. (2020b); Kurutach et al. (2018). Model exploitation can result in learning policies that perform worse than data-logging policies.

Prior works in offline model-based RL address model exploitation by enforcing conservatism for learning policies (Janner et al., 2019; Yu et al., 2020b; Kidambi et al., 2020; Yu et al., 2021) or

---

*Corr. to: Cevahir Koprulu (cevahir.koprulu@utexas.edu), Franck Djeumou (djeumf2@rpi.edu). Code

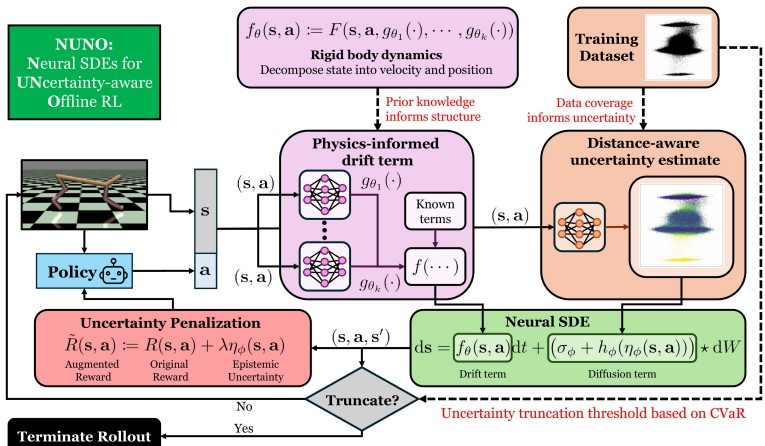

Figure 1: NUNO learns a dynamics model as neural stochastic differential equations, where its drift term can leverage prior physics knowledge as inductive bias, and its diffusion term provides distance-aware estimates of uncertainty. NUNO addresses model exploitation inherent in offline model-based RL by penalizing and adaptively truncating neural SDE's rollouts according to uncertainty estimates.

dynamics models (Rigter et al., 2022). A standard methodology of imposing conservatism is to penalize the agent with respect to the predicted uncertainty of the learned model on a taken transition Yu et al. (2020b); Kidambi et al. (2020); Yang et al. (2021); Zhang et al. (2023b). Given a true admissible error estimator for learned dynamics, these approaches provide theoretical guarantees for lower bounds on the expected cumulative reward in the groundtruth environment (Yu et al., 2020b). In practice, the standard architecture for learning dynamics models is deep probabilistic ensembles. The error estimators rely on heuristics such as maximum aleatoric uncertainty, i.e., the maximum standard deviation of learned models in the ensemble, the maximum pairwise difference between predictions of ensemble members, or variance of the log-likelihood of members. (Lu et al., 2021).

Inspired by Djeumou et al. (2023) that shows neural stochastic differential equations improve uncertainty estimates and prediction accuracy over probabilistic ensembles, we develop an offline model-based RL approach that leverages them: *Neural Stochastic Differential Equations for UNcertainty-aware, Offline RL* (NUNO; see Figure 1). NUNO learns a dynamics model as neural stochastic differential equations (SDE) and introduces

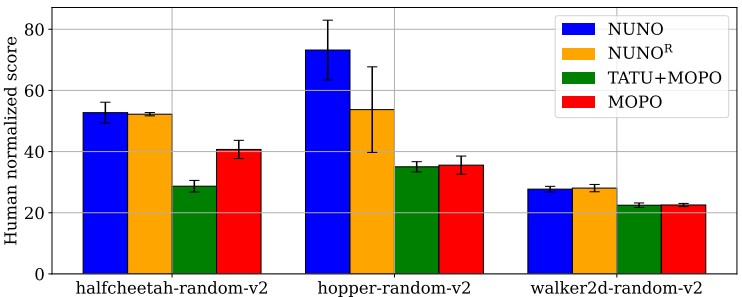

Figure 2: Comparison in random datasets of D4RL MuJoCo benchmark Fu et al. (2020). MOPO and TATU+MOPO penalize and truncate, rollouts based on uncertainty estimates from Gaussian ensembles, whereas NUNO achieves SOTA results in all environments via distance-aware uncertainty estimates of learned neural SDEs (see Fig. 1) NUNO$^R$ predicts rewards, whereas NUNO uses the groundtruth reward function.

conservatism through its uncertainty estimates. Neural SDEs consist of two main terms: drift and diffusion. A priori available physics knowledge imposes inductive biases on the drift term as a differentiable composition of separately parameterized known and unknown functions. At the same time, the diffusion term provides aleatoric and distance-aware estimate of the model uncertainty: It emulates the stochasticity of groundtruth dynamics around the training data regime while corresponding to conservative estimates of uncertainty beyond the dataset. Building on Yu et al. (2020b); Zhang et al. (2023b), NUNO addresses model exploitation by penalizing control policies and truncating training rollouts according to distance-aware uncertainty estimates of neural SDEs. NUNO provides a consistently high-performing framework, especially in randomly collected datasets (see Figure 2), by exploiting neural SDEs' capability of accurate predictions over long horizons with separate and theoretically motivated quantification of aleatoric and epistemic uncertainty.

**Contribution.** Our contribution is three-fold:

(1) We develop an uncertainty-aware offline model-based RL approach, `NUNO`, that (i) learns a dynamics model as a neural SDE, where the drift term leverages minimal prior physics knowledge as inductive bias, and the diffusion term provides distance-aware estimates of model uncertainty, and (ii) addresses model exploitation by penalizing and adaptively truncating synthetic rollouts based on a distance-aware uncertainty estimator.

(2) We theoretically show that penalization via a distance-aware uncertainty estimator enables conservatism by encouraging the policy to stay close to the convex hull of the offline data.

(3) In control benchmarks D4RL (Fu et al., 2020) and NeoRL (Qin et al., 2022), `NUNO` imposes structure on the drift term by exploiting the fact that MuJoCo environments are governed by rigid body dynamics, and decomposing state into position and velocity components.

(4) Our empirical results evidence that `NUNO` outperforms state-of-the-art methods in low-quality datasets ('random-v2' in D4RL and 'Low' in NeoRL) by up to $93\%$ while either matching or surpassing their performance by up to $55\%$ in high-quality counterparts.

## 2 RELATED WORK

Our work focuses on the intersection of offline RL and physics-informed learning of dynamics models. Appendix A investigates existing works on offline model-free RL.

**Offline model-based RL:** The objective is to learn a dynamics model from a static dataset of environment interactions in a supervised manner and subsequently generate synthetic data to train a control policy. To tackle model exploitation, most impose conservatism by constraining the learning policy to the behavior policy (Matsushima et al., 2020; Swazinna et al., 2021; Cang et al., 2021; Bhardwaj et al., 2023), learning conservative value functions (Yu et al., 2021; Rigter et al., 2022), learning pessimistic policies via biased sampling from a belief distribution over dynamics (Guo et al., 2022) or via uncertainty penalization (Yu et al., 2020b; Kidambi et al., 2020; Yang et al., 2021; Rafailov et al., 2021; Zhang et al., 2023b; Sun et al., 2023). To address compounding estimation errors, Jeong et al. (2022) propose a methodology that combines model-based and model-free value estimates for policy evaluation based on their epistemic uncertainties. A recent line of works casts offline model-based RL as a sequence modeling problem and learns a dynamics model as a transformer (Chen et al., 2021; Janner et al., 2021; Yamagata et al., 2023) without enforcing conservatism. `NUNO` inherits its principle of uncertainty penalization and rollout truncation from MOPO (Yu et al., 2020b) and TATU+MOPO (Zhang et al., 2023b), respectively. Both train a deep probabilistic ensemble as a dynamics model and penalize the reward based on their uncertainty estimator. TATU+MOPO extends MOPO by truncating synthetic trajectories if the accumulated uncertainty exceeds a threshold based on single-step estimates on the training data. `NUNO` builds on them by training a neural SDE as a dynamics model to improve uncertainty estimation and prediction accuracy (Djeumou et al., 2023).

**Neural differential equations for physics-informed learning:** Neural ordinary differential equations (ODEs) specify a structure that parameterizes a differential equation via neural networks using a priori known physics knowledge. Many existing works utilize neural ODE-based physics-informed architectures to learn dynamics models for control tasks Liu & Wang (2021); Shi et al. (2019); Plaza et al. (2022); Furieri et al. (2022); Wong et al. (2022); Menda et al. (2019); Gupta et al. (2020); Duong & Atanasov (2021); Lutter et al. (2019). Although not for control tasks, some inform the structure of neural ODEs via Hamiltonian Greydanus et al. (2019); Chen et al. (2019); Zhu et al. (2020); Zhong et al. (2020); Eidnes et al. (2023), Port-Hamiltonian Desai et al. (2021); Neary & Topcu (2023), or Lagrangian Roehrl et al. (2020); Finzi et al. (2020); Cranmer et al. (2020); Allen-Blanchette et al. (2020); Zhong et al. (2021b;a) formulation of dynamics. Neural ODE-based structures are commonly deterministic and, hence do not provide a notion of uncertainty. In contrast, neural SDEs allow uncertainty-aware models, and previous works investigate their use for learning dynamics of stochastic systems (Jia & Benson, 2019; Yang et al., 2023), estimating the uncertainty in parameters of neural networks Kong et al. (2020); Li et al. (2020); Kidger (2022); Xu et al. (2022), and generative modeling Kidger et al. (2021). However, these approaches do not deal with epistemic uncertainty in a way tailored to offline RL. Djeumou et al. (2023) propose using neural SDEs to leverage a priori physics knowledge and capture epistemic uncertainty to control dynamical systems and offline model-based RL. In comparison to our work, Djeumou et al. (2023) does not address the model exploitation problem and does not use a loss function that properly models aleatoric uncertainty.

## 3 PRELIMINARIES

### 3.1 MARKOV DECISION PROCESSES

We formalize the environments of interest in this work as Markov decision processes (MDP), specified by a tuple $\mathcal{M} = \langle \mathcal{S}, \mathcal{A}, T, R, \gamma, \rho_0 \rangle$, where $\mathcal{S}$ and $\mathcal{A}$ are state and action spaces, $T : \mathcal{S} \times \mathcal{A} \to \Delta(\mathcal{S})$ is the transition distribution, $R : \mathcal{S} \times \mathcal{A} \to \mathbb{R}$, $\gamma \in (0, 1)$ is the discount factor, and $\rho_0$ is the initial state distribution, i.e., $\Delta(\mathcal{S})$. A policy $\pi : \mathcal{S} \to \Delta(\mathcal{A})$ in an MDP $\mathcal{M}$ outputs a probability simplex over the action space $\mathcal{A}$ given a state $\mathbf{s} \in \mathcal{S}$. The objective of RL is to learn an optimal policy $\pi^*$, which maximizes the expected discounted return in $\mathcal{M}$, i.e., $\pi^* \in \arg\max_\pi \eta_\mathcal{M}(\pi)$, where $\eta_\mathcal{M}(\pi) = \mathbb{E}_{\pi, T, \rho_0}[\sum_{t=0}^\infty \gamma^t R(\mathbf{s}_t, \mathbf{a}_t)]$ is the expected discounted return, $\mathbf{a}_t \sim \pi(\mathbf{s}_t)$ is the policy's action, and $\mathbf{s}_{t+1} \sim T(\mathbf{s}_t, \mathbf{a}_t)$ is the new state at time $t$ by starting in $\mathbf{s}_0 \sim \rho_0$.

### 3.2 OFFLINE MODEL-BASED RL

The offline RL problem assumes access to a dataset $\mathcal{D} = \{\tau_i\}_i$ of interactions $\tau = \{(\mathbf{s}, \mathbf{a}, r, \mathbf{s}')_t\}_t$ with the environment $\mathcal{M}$. Multiple behavior policies $\pi_\mathrm{b}$, optimal or suboptimal, can contribute to $\mathcal{D}$. The objective is to learn a policy $\pi$ that minimizes the sub-optimality gap, namely, $\eta_\mathcal{M}(\pi^*) - \eta_\mathcal{M}(\pi)$. Offline model-based RL methods approach this problem by first learning a dynamics model $\hat{T}$ from the dataset $\mathcal{D}$. Then, they utilize the learned dynamics model to optimize the policy. Depending on the access, one can learn a reward function $\hat{R}$ or an initial state distribution $\rho_0$ from the dataset $\mathcal{D}$.

A naive way to optimize the policy is to interact with the learned MDP, e.g., as online RL algorithms. However, such an approach can cause model exploitation, i.e., the estimated return in the learned environment is greater than the true return: $\eta_{\hat{\mathcal{M}}}(\pi) - \eta_\mathcal{M}(\pi) > 0$ (Yu et al., 2020b). Due to the finite coverage of the dataset $\mathcal{D}$, the policy $\pi$ learns to exploit regions of the state-action space where the epistemic uncertainty of the learned model $\hat{T}$ and the estimated return $\eta_{\hat{\mathcal{M}}}(\pi)$ are high.

A common strategy to undertake model exploitation is to penalize the agent in correlation to the estimated model uncertainty, as in Model-based Offline Policy Optimization (MOPO) (Yu et al., 2020b), which defines a pessimistic reward function: $\tilde{R}(\mathbf{s}, \mathbf{a}) \doteq R(\mathbf{s}, \mathbf{a}) - \lambda_\mathrm{pen} u(\mathbf{s}, \mathbf{a})$, where $u(\mathbf{s}, \mathbf{a})$ is the estimation of the model uncertainty at the state-action pair $(\mathbf{s}, \mathbf{a})$ and $\lambda_\mathrm{pen}$ is the regularization coefficient for the uncertainty penalty. Utilizing the pessimistic reward function, MOPO constructs a pessimistic learned MDP $\tilde{\mathcal{M}} = \langle \mathcal{S}, \mathcal{A}, \hat{T}, \tilde{R}, \gamma, \rho_0 \rangle$ and modifies the policy optimization objective as $\max_\pi \eta_{\tilde{\mathcal{M}}}(\pi)$. Zhang et al. (2023b) proposes Trajectory Truncation with Uncertainty, TATU, which truncates model rollouts if the accumulated uncertainty exceeds a predetermined threshold. The theoretical results follow a similar line of argument in MOPO and construct a pessimistic MDP, then provide suboptimality bounds for policies learned in pessimistic MDPs. TATU's uncertainty truncation threshold depends on uncertainty estimates over the single-step transitions from the datasets, which is a limitation considering that TATU trains policies with longer rollouts.

### 3.3 NEURAL STOCHASTIC DIFFERENTIAL EQUATIONS AS DYNAMICS MODELS

Stochastic differential equations (SDEs) offer a principled approach to modeling uncertain, real-world, and time-varying stochastic processes. Their continuous-time nature and ability to encode prior physics knowledge (world models) as inductive bias make them suitable for modeling dynamical systems from data. A neural SDE is an SDE parameterized by neural networks as follows

$$d\mathbf{s} = f_\theta(\mathbf{s}, \mathbf{a})\, dt + \Sigma_\phi(\mathbf{s}, \mathbf{a}) \star dW, \tag{1}$$

where $f_\theta : \mathcal{S} \times \mathcal{A} \to \mathbb{R}^{n_\mathbf{s}}$ and $\Sigma_\phi : \mathcal{S} \times \mathcal{A} \to \mathbb{R}^{n_\mathbf{s} \times n_w}$ are the drift and diffusion terms parameterized by $\theta$ and $\phi$, $W$ is the $n_w$-dimensional Wiener process, and $\star$ expresses that the SDE is either in Ito Ito et al. (1951) or Stratonovich Stratonovich (1966) form. The reader unfamiliar with these forms should feel free to ignore the distinction (Van Kampen, 1981; Massaroli et al., 2021; Kidger, 2022), which becomes an arbitrary modeling choice when $f_\theta$ and $\Sigma_\phi$ are learned.

Given a dataset $\mathcal{D}$ of interactions of a behavior policy $\pi_\mathrm{b}$ in MDP $\mathcal{M}$, we seek the unknown functions $f_\theta$ and $\Sigma_\phi$ of a neural SDE that best fit the sequences of states and actions in the dataset. Specifically, we build on the framework proposed in Djeumou et al. (2023) and extend it to train neural SDEs in such a way that the diffusion term $\Sigma_\phi$ captures aleatoric uncertainty as well as epistemic uncertainty in the form distance-aware estimates of model uncertainty.

## 4 NUNO

In this section, we discuss the details of NUNO's design. First, we provide insight into the use of a distance-aware uncertainty estimator and discuss our parametric estimator and its corresponding training algorithm. Then, we introduce our physics-based neural SDE approach for modeling the dynamics of the MDP while capturing aleatoric and epistemic uncertainty. Finally, we show how our uncertainty estimator can efficiently enforce conservatism when training the RL policy. In the remainder of the paper, we assume access to the dataset $\mathcal{D}$ of realized state-actions trajectories. We also assume access to the time steps $\Delta t$ between consecutive states $\mathbf{s}_t$ and $\mathbf{s}_{t+1}$.

### 4.1 DISTANCE-AWARE UNCERTAINTY ESTIMATOR

By investigating particle-based estimators of the cross entropy between the learned model's transition distribution and the unknown transition distribution, Zhang et al. (2023a) provides a theoretical framework for characterizing model uncertainty $u(\mathbf{s}, \mathbf{a})$ as a function of the distance, in the appropriate space, between the query point $(\mathbf{s}, \mathbf{a})$ and *its $k$-th nearest neighbor (KNN) in the dataset $\mathcal{D}$*. We build on this idea and propose a parametric distance-aware uncertainty estimator $\eta_\phi : \mathcal{S} \times \mathcal{A} \to \mathbb{R}$ that captures such distance to the closest $k$-th neighbor in the dataset without the need for a KNN search. Besides bypassing intractable KNN search, our parametric estimator can be trained alongside the neural SDE model (see Section 4.2) such that the model can capture both aleatoric and epistemic uncertainty in the dynamics. The estimator is smooth and differentiable and thus blends well with the requirements for numerical integration of the neural SDE model.

A simple choice for $\eta_\phi$ for which we can provide theoretical guarantees is given by

$$\bar{\eta}_\phi = \mathrm{argmin}_\eta \, \mathbb{E}_{(\mathbf{s},\mathbf{a}) \sim \mathcal{D}} \big[ \mathbb{E}_{(\mathbf{s}',\mathbf{a}') \sim \mathrm{Uniform}(\mathcal{S} \times \mathcal{A})} [\eta(\mathbf{s}', \mathbf{a}') - \|(\mathbf{s}, \mathbf{a}) - (\mathbf{s}', \mathbf{a}')\|]^2 \big]. \quad (2)$$

**Lemma 1.** *The optimal solution $\bar{\eta}_\phi$ of equation 2 is a convex function with respect to $(\mathbf{s}, \mathbf{a})$ and is an upper bound of the distance to the state-action centroid of the training dataset. Additionally, we have that the negative gradient $-\nabla_{\mathbf{s},\mathbf{a}} \bar{\eta}_\phi$ at any point $(\mathbf{s}, \mathbf{a})$ points inside the convex hull of $\mathcal{D}$.*

We provide the proof of Lemma 1 in Appendix B. The first property above illustrates that $\bar{\eta}_\phi$ is a suitable choice for a distance-aware uncertainty estimator. In contrast, the second property enables conservatism by suggesting that any reward penalization with $\bar{\eta}_\phi$ will encourage the policy to stay within the convex hull of the training dataset. However, the estimator $\bar{\eta}_\phi$ approximates only the distance to the centroid of the entire dataset, which may not be sufficient to accurately capture the uncertainty in the model's predictions if the geometry of the dataset has multiple clusters.

To address this limitation, we enforce additional constraints to encourage $\eta_\phi$ to cluster the dataset properly. Informally speaking, we model the term $\eta_\phi(\cdot)$ with neural networks such that when evaluated near points in the training dataset $\mathcal{D}$, such term provides low values with almost-zero gradients. In contrast, it provides high but bounded values when evaluated far from the training data. Specifically, a strong property of our approach is that by sampling only locally around the training dataset, we can train the parameters of $\eta_\phi$ to enforce the desired distance-based properties globally. In particular, we translate the distance-aware requirement into several mathematical properties that $\eta_\phi$ must satisfy, and propose a loss function that encourages the neural network to learn these properties.

(a) *Increasing $\eta_\phi$ along state-action paths that move away from the training data.* As the query point $(\mathbf{s}, \mathbf{a})$ moves away from the training data, the distance-aware term $\eta_\phi$ should monotonically increase accordingly. Let $\Gamma$ be any path along which the distance from the current point to the nearest training datapoint always increases. Then, along $\Gamma$, the entries of $\eta_\phi$ should monotonically increase. We enforce this property via local strong convexity constraints near the training dataset. Specifically, for every state action $(\mathbf{s}_t, \mathbf{a}_t) \in \mathcal{D}$ and a fixed radius $r > 0$, we enforce strong convexity of $\eta_\phi$ within a ball $\mathcal{B}_r(\mathbf{s}_t, \mathbf{a}_t) := \{(\mathbf{s}, \mathbf{a}) \mid \|(\mathbf{s}, \mathbf{a}) - (\mathbf{s}_t, \mathbf{a}_t)\| \leq r\}$ with a convexity constant $\mu_t > 0$. More specifically, we want to enforce that for any $(\mathbf{s}, \mathbf{a}), (\mathbf{s}', \mathbf{a}') \in \mathcal{B}_r(\mathbf{s}_t, \mathbf{a}_t)$, the convexity constraint $(\mathbf{s}, \mathbf{a}, \mathbf{s}', \mathbf{a}')_{\mu_t} \geq 0$ holds, where the constraint is defined as

$$(\mathbf{s}, \mathbf{a}, \mathbf{s}', \mathbf{a}')_{\mu_t} := \eta_\phi(\mathbf{s}', \mathbf{a}') - \eta_\phi(\mathbf{s}, \mathbf{a}) - \nabla_{(\mathbf{s},\mathbf{a})} \eta_\phi(\mathbf{s}, \mathbf{a})^\top ((\mathbf{s}, \mathbf{a}) - (\mathbf{s}', \mathbf{a}')) - \mu_t \|(\mathbf{s}, \mathbf{a}) - (\mathbf{s}', \mathbf{a}')\|^2.$$

We parametrize a function $\mu_\phi : \mathcal{S} \times \mathcal{A} \to \mathbb{R}_+$ using a neural network to predict the strong convexity constants $\mu_t = \mu_\phi(\mathbf{s}_t, \mathbf{a}_t)$ for each $(\mathbf{s}_t, \mathbf{a}_t) \in \mathcal{D}$ instead of manually tuning them. Thus, we define

the following loss functions to enforce the desired properties at a sample $(\mathbf{s}_t, \mathbf{a}_t) \in \mathcal{D}$ as follows:

$$\mathcal{L}_{\text{sc}} = \sum_{\substack{(\mathbf{s},\mathbf{a}),(\mathbf{s}',\mathbf{a}') \\ \sim \mathcal{N}((\mathbf{s}_t,\mathbf{a}_t),r)}} \begin{cases} 0, \text{ if } (\mathbf{s},\mathbf{a},\mathbf{s}',\mathbf{a}')_{\mu_t} \geq 0 \\ (\mathbf{s},\mathbf{a},\mathbf{s}',\mathbf{a}')^2_{\mu_t}, \text{ otherwise} \end{cases} \quad \text{and } \mathcal{L}_{\mu} = \sum_{(\mathbf{s}_t,\mathbf{a}_t) \in \mathcal{D}} \frac{1}{\mu_\phi(\mathbf{s}_t,\mathbf{a}_t)}, \quad (3)$$

where $\mathcal{N}((\mathbf{s}_t, \mathbf{a}_t), r)$ is a Gaussian distribution with mean $(\mathbf{s}_t, \mathbf{a}_t)$ and standard deviation $r$, and $\mathcal{L}_\mu$ is a regularization loss term that encourages high values of $\mu_t$. Intuitively, such regularization ensures that the distance-aware term $\eta_\phi$ reaches its maximum value as close as possible to the boundaries of the training dataset, *enabling dataset clustering*.

(b) *Zero-gradient and distance-aware estimate near training data.* We enforce that the distance-aware term $\eta_\phi$ has almost zero gradients near the training dataset such that, with the local convexity constraints, points in the dataset become local minima of $\eta_\phi$ and the negative gradient of $\eta_\phi$ near a cluster is directed towards the cluster. This constraint can be enforced at a sample $(\mathbf{s}_t, \mathbf{a}_t) \in \mathcal{D}$ as

$$\mathcal{L}_{\text{grad}} = \|\nabla_{(\mathbf{s},\mathbf{a})}\eta_\phi(\mathbf{s}_t,\mathbf{a}_t)\|^2 + \eta_\phi(\mathbf{s}_t,\mathbf{a}_t)^2, \quad (4)$$

where the last term encourages $\eta_\phi$ to be zero when evaluated on the training data. Appendix B.2 provides insights about the distance-aware uncertainty estimator, as well as toy 2-D dataset examples to demonstrate how $\eta_\phi$ efficiently clusters the training dataset to capture datapoints distance.

## 4.2 Physics-Inspired Neural SDEs

We aim to learn a neural SDE's drift and diffusion terms that best fit the sequences of states and actions in the dataset $\mathcal{D}$. Specifically, we first consider the following black-box neural SDE

$$d\mathbf{s} = f_\theta(\mathbf{s}, \mathbf{a}) \, dt + \left(\sigma_\phi(\mathbf{s}, \mathbf{a}) + h_\phi(\eta_\phi(\mathbf{s}, \mathbf{a}))\right) \star dW, \quad (5)$$

where we simplify the diffusion term $\Sigma_\phi$ from equation 1 to be a diagonal matrix composed of two complementary terms. The first term $\sigma_\phi : \mathcal{S} \times \mathcal{A} \to \mathbb{R}^{n_{\mathbf{s}}}$ is an unconstrained neural network that captures the aleatoric uncertainty of the dynamics, while we design the second term $h_\phi(\eta_\phi(\cdot))$ to estimate heterogeneous epistemic uncertainty in the model's predictions. Here $h_\phi : \mathbb{R} \to \mathbb{R}^{n_{\mathbf{s}}}$ is a bounded, monotonic, and learnable transformation that ensures the diffusion term is positive and monotonically increasing in the proposed distance-aware term $\eta_\phi : \mathcal{S} \times \mathcal{A} \to \mathbb{R}$. In the following, we use $\Sigma_\phi$ to refer to $\sigma_\phi + h_\phi(\eta_\phi)$ when the distinction is unnecessary.

**Monotonicity and boundedness of $h_\phi$.** To ensure globally monotonic and bounded diffusion values as a function of $\eta_\phi$, we adopt a simple design choice for $h_\phi$: A scaled sigmoid function to transform $\eta_\phi$ into a heterogeneous diffusion term. Specifically, we define $h_\phi(\eta_\phi) = W^{\text{max}}\text{sigmoid}(W\eta_\phi + b)$, where $\text{sigmoid}(x) = (1 + \exp(-x))^{-1}$, $W \in \mathbb{R}^{n_{\mathbf{s}}}$ and $b \in \mathbb{R}^{n_{\mathbf{s}}}$ are the learnable parameters of the neural network. We constrain $W$ to be greater than 1. Besides, the term $W^{\text{max}}$ is a hyperparameter that controls the desired maximum value of the diffusion term outside the training data regime. We emphasize that this design choice works well in our experiments, but others are possible.

**Training the neural SDE.** In contrast to the standard approaches such as probabilistic ensembles where the model fits a single-step transition, the proposed neural SDE is designed and trained to fit sequences of states and the uncertainty in the model's predictions. Given a sequence of states and actions $\{\mathbf{s}_t, \mathbf{a}_t, \ldots, \mathbf{s}_{t+H}\}$ with H being the prediction horizon, we aim to minimize the negative log-likelihood (NLL) of the sequence under the neural SDE model. However, estimating the NLL of neural SDE-generated sequences is challenging due to the intractability of computing the kernel density of the underlying stochastic process. To address this issue, we adopt numerical integration schemes to approximate the sequence's NLL through Monte Carlo sampling. Specifically, assuming approximate Gaussian transitions between discrete time steps of the stochastic process, e.g., when employing the Euler-Maruyama sampler, we can approximate the NLL as

$$\mathcal{L}_{\text{data}} = \mathbb{E}_{\tilde{\mathbf{s}}_{t+1}^{\theta,\phi}, \ldots, \tilde{\mathbf{s}}_{t+H}^{\theta,\phi}} \left[ \sum_{k=t}^{t+H-1} \|\mathbf{s}_{k+1} - \tilde{\mathbf{s}}_{k+1}^{\theta,\phi}\|^2_{(\Sigma_\phi^{-1})_k} + \log(\det(\Sigma_\phi)_k) \right], \quad (6)$$

where $(\Sigma_\phi)_k = \Sigma_\phi(\tilde{\mathbf{s}}_k^{\theta,\phi}, \mathbf{a}_k)$, and $\tilde{\mathbf{s}}_{t+1}^{\theta,\phi}, \ldots, \tilde{\mathbf{s}}_{t+H}^{\theta,\phi}$ are the sample states obtained by any differential SDE numerical integration scheme. Note that the accuracy of the NLL approximation depends on the quality of the numerical integration scheme. the stepsize to discretize the SDE between two

consecutive states, and the number of samples used to estimate the expectation. In practice, though, we can fit accurate neural SDE models to the data even with Euler-Maruyama and a single sample.

The problem of learning the SDE model parameters with distance-aware uncertainty estimates can be formulated as the following optimization problem:

$$\underset{\theta,\phi}{\text{minimize}} \quad \mathbb{E}_{\mathbf{s}_t,\mathbf{a}_t,\ldots,\mathbf{s}_{t+\mathrm{H}} \sim \mathcal{D}} \left[ \lambda_{\mathrm{data}} \mathcal{L}_{\mathrm{data}} + \lambda_{\mathrm{sc}} \mathcal{L}_{\mathrm{sc}} + \lambda_{\mathrm{grad}} \mathcal{L}_{\mathrm{grad}} + \lambda_{\mu} \mathcal{L}_{\mu} \right], \quad (7)$$

**Incorporating prior physics knowledge.** We can incorporate prior physics knowledge into the neural SDE model by designing the drift term $f_\theta$ to encode structural knowledge from first principles or domain expertise. To this end, we represent the drift term $f_\theta$ as the composition of a known function – derived from a priori knowledge – and a collection of unknown functions that must be learned from data. That is, we write $f_\theta(\mathbf{s}, \mathbf{a}) := F(\mathbf{s}, \mathbf{a}, g_{\theta_1}(\cdot), \ldots, g_{\theta d}(\cdot))$, where $F$ is a known differentiable function and $g_{\theta_1}(\cdot), \ldots, g_{\theta d}(\cdot)$ are unknown terms within the underlying model. The inputs to these functions could themselves be arbitrary functions of the states and control inputs. Additionally, known constraints on $g_{\theta_i}$ can be enforced during training using the augmented Lagrangian method.

We exploit the fact that rigid body dynamics govern our benchmark environments to constrain the structure on the drift term. We typically decompose the state as $\mathbf{s} = [\mathbf{s}_{\mathrm{pos}}, \mathbf{s}_{\mathrm{vel}}]$, where $\mathbf{s}_{\mathrm{pos}}$ and $\mathbf{s}_{\mathrm{vel}}$ are the position and velocity components, respectively, and we define the drift term as

$$f_\theta^{\mathrm{pos}}(\mathbf{s}, \mathbf{a}) = \mathbf{s}_{\mathrm{vel}}, \ f_\theta^{\mathrm{vel}}(\mathbf{s}, \mathbf{a}) = G_\phi(\mathbf{s}_{\mathrm{vel}})\mathbf{a} + H_\phi(\mathbf{s})\mathbf{s}_{\mathrm{vel}}, \ f_\theta = [f_\theta^{\mathrm{pos}}, f_\theta^{\mathrm{vel}}] + f_\theta^{\mathrm{res}}, \quad (8)$$

where $G_\phi$ and $H_\phi$ are learnable neural networks, and $f_\theta^{\mathrm{res}}$ is a residual term capturing unmodeled dynamics. We penalize the residual term to ensure minimal deviation from the drift term. Note that this formulation integrates minimal prior knowledge into the neural SDE model, and such knowledge does not affect modeling performance in the large dataset regime seen in our experiments. For a discussion on general prior physics knowledge, we refer the reader to Djeumou et al. (2023).

**Incorporating reward learning.** We can incorporate reward learning into the neural SDE model by augmenting the state representation with a variable representing cumulative rewards. Specifically, we define the new state as $\mathbf{s} = [\mathbf{s}_{\mathrm{pos}}, \mathbf{s}_{\mathrm{vel}}, r_{\mathrm{c}}]$, where $r_{\mathrm{c}}$ is the cumulative reward up to the current time step. We then augment the neural SDE model with $\mathrm{d}r_{\mathrm{c}} = f_\theta^{\mathrm{rew}}(\mathbf{s}, \mathbf{a})\,\mathrm{d}t$, where $f_\theta^{\mathrm{rew}}$ is a learnable neural network that captures the reward dynamics. We can then train the combined neural SDE model to minimize the NLL of the sequence of states, actions, and rewards under the model.

## 4.3 Distance-Aware Regularized Offline RL

We now discuss incorporating the distance-aware uncertainty estimate $\eta_\phi$ into the offline RL framework to enforce conservatism in the learned policy. Specifically, we build on the work by Zhang et al. (2023b) and use our distance-aware uncertainty estimate to penalize and truncate the transitions generated by the learned neural SDE model during the RL policy training.

**Reward penalty.** Following MOPO penalization criteria, we use the distance-aware uncertainty to define the pessimistic reward as $\tilde{R}(\mathbf{s}, \mathbf{a}) = R(\mathbf{s}, \mathbf{a}) - \lambda_{\mathrm{pen}}\eta_\phi(\mathbf{s}, \mathbf{a})$.

**Trajectory truncation.** During the RL agent training, we use the current policy and the neural SDE model to generate synthetic trajectories for policy improvement. To figure out whether the synthetic trajectory is reliable, we set a truncating threshold $\epsilon$ on the accumulated distance-aware estimate $\eta_\phi$ over the sequence. Specifically, we compute $\mathcal{T} = \sum_{t=0}^{h} \eta_\phi(\mathbf{s}_t, \mathbf{a}_t)$, and we compare its value with the threshold $\epsilon$. If the accumulated quantity exceeds the threshold, we truncate the trajectory and do not use it for policy optimization. The choice of the threshold $\epsilon$ is a crucial hyperparameter that varies accross environments or tasks while enforcing the level of conservatism in the policy training. To account for different task and environment complexities, we propose automatically setting the threshold based on the entire training dataset. We propose to use a user-defined Conditional Value-at-Risk (CVaR) as the threshold to compute the hyperparameter $\epsilon$ via performing statistics on the entire dataset over all possible sequences of horizon $h$.

## 5 Experimental Results

We empirically evaluate NUNO against state-of-the-art (SOTA) offline model-based and model-free approaches in continuous control benchmarks, namely MuJoCo datasets in D4RL Fu et al. (2020)

Table 1: Average human-normalized scores of NUNO and other model-based and model-free offline RL approaches on D4RL MuJoCo v2 datasets. Due to limited space, we use abbreviations of task and dataset names: hc = halfcheetah, hp = hopper, wk = walker2d; r = random, m = medium, mr = medium-replay, me = medium-expert. For NUNO, we provide the mean and standard deviation (following $\pm$) of best scores among independent runs. Bold scores indicate the best for each task.

| Task | NUNO (Ours) | NUNO$^R$ (Ours) | MOBILE | MOPO$^T$ | MOPO | COMBO | MOReL | RAMBO | EDAC |
|---|---|---|---|---|---|---|---|---|---|
| hc-r | **52.7±3.4** | **52.2±0.5** | 39.3±3.0 | 33.3 | 35.9 | 38.8 | 38.9 | 39.5 | 28.4 |
| hp-r | **73.2±9.8** | **53.7±13.9** | 31.9±0.6 | 31.9 | 16.7 | 17.9 | 38.1 | 25.4 | 25.3 |
| wk-r | **27.7±0.9** | **28.1±1.2** | 17.9±6.6 | 10.4 | 4.2 | 7.0 | 16.0 | 0.0 | 16.6 |
| hc-m | 68.8±0.4 | 64.7±0.5 | 74.6±1.2 | 61.9 | 73.1 | 54.2 | 60.7 | **77.9** | 65.9 |
| hp-m | 104.6±0.2 | 104.4±0.3 | **106.6±0.6** | 104.3 | 38.3 | 97.2 | 84.0 | 87.0 | 101.6 |
| wk-m | 85.4±0.9 | **92.6±1.3** | 87.7±1.1 | 77.9 | 41.2 | 81.9 | 72.8 | 84.9 | **92.5** |
| hc-mr | 66.5±0.2 | 64.6±0.3 | **71.7±1.2** | 67.2 | 69.2 | 55.1 | 44.5 | 68.7 | 61.3 |
| hp-mr | **107.8±1.2** | 106.6±1.9 | 103.9±1.0 | 104.4 | 32.7 | 89.5 | 81.8 | 99.5 | 101.0 |
| wk-mr | **97.0±1.4** | **101.1±3.9** | 89.9±1.5 | 75.3 | 73.7 | 56.0 | 40.8 | 89.2 | 87.1 |
| hc-me | 97.0±0.5 | 95.8±1.2 | **108.2±2.5** | 74.1 | 70.3 | 90.0 | 80.4 | 95.4 | **106.3** |
| hp-me | **112.2±0.3** | 111.9±0.5 | **112.6±0.2** | 107.0 | 60.6 | 111.1 | 105.6 | 88.2 | 110.7 |
| wk-me | 113.2±0.5 | 112.6±0.6 | **115.2±0.7** | 107.9 | 77.4 | 103.3 | 107.5 | 56.7 | **114.7** |
| Average | **83.8** | 82.4 | 80.0 | 71.3 | 49.4 | 66.8 | 64.3 | 67.7 | 76.0 |

and NeoRL Qin et al. (2022). Through our empirical evaluation, we answer the following questions: 1) How does NUNO perform in terms of human normalized score? 2) Can NUNO's uncertainty estimator, i.e., distance-aware estimate of a neural SDE, effectively quantify uncertainty? 3) How does NUNO address the model exploitation phenomenon in contrast to TATU+MOPO and MOPO?

## 5.1 HOW DOES NUNO PERFORM IN STANDARD CONTROL BENCHMARKS?

### 5.1.1 D4RL

We run experiments on 12 D4RL tasks, combining three MuJoCo environments (halfcheetah, hopper, and walker2d) and four datasets (random, medium, medium-replay, and medium-expert) per environment. We compare NUNO against a model-free method called EDAC An et al. (2021), that penalizes Q-values based on the estimated uncertainty of a Q-function ensemble; and model-based methods: MOPO Yu et al. (2020b) and TATU+MOPO Zhang et al. (2023b), from which NUNO inherits uncertainty penalization and truncation, respectively, COMBO Kumar et al. (2020), which equally penalizes samples that are out-of-distribution according to model uncertainty, MOBILE Sun et al. (2023), which penalizes the Bellman estimation based on the inconsistency of Bellman estimations by an ensemble of learned dynamics models, RAMBO that adversarially learns a policy and dynamics model, and finally, MOReL Kidambi et al. (2020), which penalizes a transition when estimated uncertainty exceeds a threshold. In Table 1, we refer to TATU+MOPO as MOPO$^T$.

Table 1 demonstrates the mean and standard deviation of maximum human-normalized scores that NUNO and NUNO$^R$, which predicts the reward, reach in D4RL MuJoCo tasks (v2) during five independent runs of one million gradient steps. In the random task involving datasets collected by randomly initialized policies, NUNO and NUNO$^R$ outperform all approaches by a significant margin. NUNO achieves this by building onto uncertainty penalization and truncation principles proposed by MOPO and TATU+MOPO. NUNO's advantage comes from leveraging minimal prior physics knowledge and exploiting the diffusion term's capability of estimating aleatoric and epistemic uncertainty. Given higher-quality datasets, namely, better-performing data logging policies, NUNO either reaches SOTA results or closely follows existing ones. Specifically, NUNO reaches state-of-the-art results in hopper-medium-replay-v2, walker-medium-replay-v2, and hopper-medium-expert-v2. Table 1 also demonstrates that NUNO achieves the second-best results in medium tasks of halfcheetah and hopper, as well as medium-expert task of hopper. Overall, NUNO and NUNO$^R$ yield the highest average human-normalized scores in the D4RL MuJoCo benchmark. Figure 3a visualize the progression of human normalized score for NUNO and NUNO$^R$. See Appendix E.2 for more details.

Table 2: Average human-normalized scores of NUNO and other model-based and model-free offline RL approaches on NeoRL MuJoCo datasets. Due to limited space, we use abbreviations of dataset names: L = low, M = medium, H = high. For NUNO, we provide the mean and standard deviation (following ±) of best scores among independent runs. Bold scores indicate the best for each task.

| Task | NUNO (Ours) | NUNO[R] (Ours) | MOBILE | MOPO | BC | CQL | TD3+BC | EDAC |
|------|-------------|----------------|--------|------|------|------|--------|------|
| hc-L | 52.5±0.6 | **58.4±0.5** | 54.7±3.0 | 40.1 | 29.1 | 38.2 | 30.0 | 31.3 |
| hp-L | **26.9±3.8** | **26.4±6.8** | 17.4±3.9 | 6.2 | 15.1 | 16.0 | 15.8 | 18.3 |
| wk-L | **52.5±2.4** | 49.4±1.9 | 37.6±2.0 | 11.6 | 28.5 | 44.7 | 43.0 | 40.2 |
| hc-M | 73.4±0.6 | **78.8±0.8** | 77.8±1.4 | 62.3 | 49.0 | 54.6 | 52.3 | 54.9 |
| hp-M | **103.3±2.2** | 92.3±1.7 | 51.1±13.3 | 1.0 | 51.3 | 64.5 | 70.3 | 44.9 |
| wk-M | **65.8±0.4** | 49.4±16.9 | 62.2±1.6 | 39.9 | 48.7 | 57.3 | 58.5 | 57.6 |
| hc-H | **85.2±0.6** | 84.9±0.4 | 83.0±4.6 | 65.9 | 71.3 | 77.4 | 75.3 | 81.4 |
| hp-H | **103.0±3.1** | 97.9±5.5 | 87.8±26.0 | 11.5 | 43.1 | 76.6 | 75.3 | 52.5 |
| wk-H | 72.9±1.6 | 74.5±1.6 | 74.9±3.4 | 18.0 | 72.6 | 75.3 | 69.6 | **75.5** |
| Average | **70.6** | 68 | 60.7 | 28.5 | 45.4 | 56.1 | 54.5 | 50.7 |

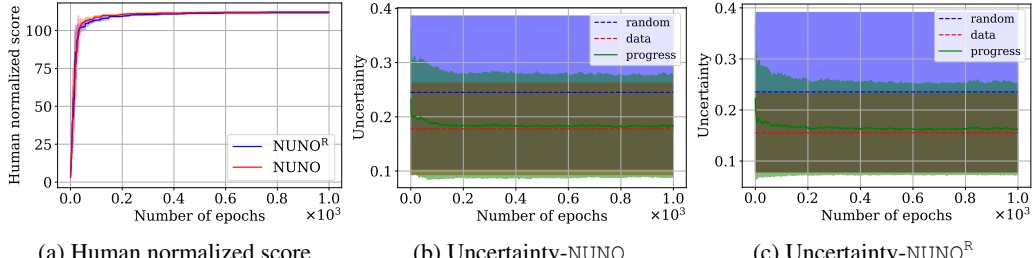

(a) Human normalized score      (b) Uncertainty-NUNO      (c) Uncertainty-NUNO[R]

Figure 3: Training progression in hopper-medium-expert-v2: (a) We report the progression of human normalized score in evaluation episodes during training. (b-c) We demonstrate how the uncertainty estimates of neural SDEs in NUNO and NUNO[R] evolve when evaluated with trained policies' actions in one-step rollouts from states in the dataset. 'random' and 'data' refer to the uncertainty estimates of the learned model given actions from a random policy and the dataset, respectively.

### 5.1.2 NEORL

We further evaluate NUNO in NeoRL Qin et al. (2022), a benchmark developed to reflect real-world characteristics by logging data via conservative policies. We investigate nine datasets involving three environments (HalfCheetah-v3, Hopper-v3, Walker2d-v3) and three types of datasets (low, medium, high) per environment with 1000 trajectories each. We compare NUNO against MOBILE, MOPO, EDAC, CQL which penalizes OOD samples' Q-values equally, behavior cloning (BC), which imitates data-logging policies, and TD3+BC Fujimoto & Gu (2021), which extends TD3 Fujimoto et al. (2018) by regularizing the policy optimization objective via a behavioral cloning term.

Table 2 reports the mean and standard deviation of maximum human-normalized scores that NUNO reaches in NeoRL MuJoCo tasks during four independent runs of one million gradient steps. NUNO achieves the highest scores in the low tasks of NeoRL by outperforming existing SOTA results in hopper and walker2d by a significant margin, as in the random tasks of D4RL. In addition, NUNO or NUNO[R] reach the highest scores in medium and high tasks of all MuJoCo environments. Overall, NUNO and NUNO[R] collect the highest average human normalized scores across nine tasks in NeoRL.

### 5.2 CAN NUNO'S UNCERTAINTY ESTIMATOR EFFECTIVELY QUANTIFY UNCERTAINTY?

Figures 3b and 3c demonstrate the evolution of uncertainty estimates of trained neural SDEs for NUNO and NUNO[R] during training. Neural SDE's uncertainty estimator assigns the largest values to random actions and the smallest to the dataset, evidencing that the uncertainty estimators correctly identify out-of-distribution and in-distribution actions, respectively. As trained policies progress, see Figure 3a, the model uncertainty for learned policies' actions approaches the uncertainty of in-distribution samples because learned policies avoid out-of-distribution actions through penalization and truncation based on distance-aware uncertainty estimates. Appendix D provides an ablation study on the choice of the uncertainty estimator for penalization and truncation in policy learning.

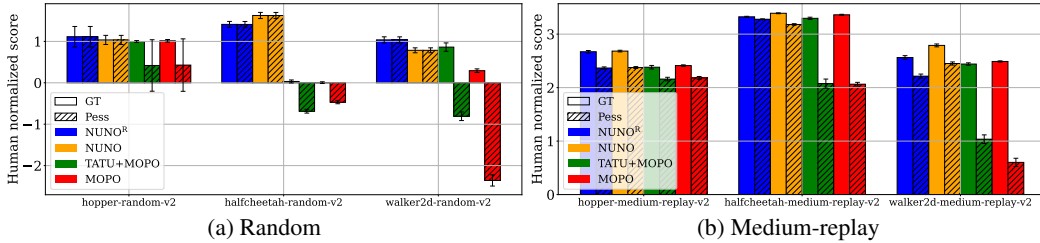

|     |     |
| :-: | :-: |
| (a) Random | (b) Medium-replay |

Figure 4: Model exploitation: We evaluate NUNO, NUNO[R], TATU+MOPO, and MOPO in rollouts from their learned dynamics models in (a) random and (b) medium-replay tasks, and report the average score per step with (pessimistic, Pess) and without (groundtruth, GT) uncertainty penalization.

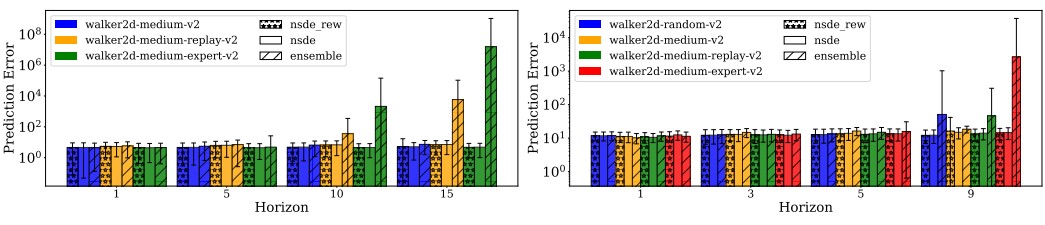

|     |     |
| :-: | :-: |
| (a) D4RL Walker2d: In-distribution | (b) D4RL Walker2d: Out-of-distribution |

Figure 5: Model analysis: We illustrate the evolution of model prediction error in different datasets for D4RL Walker2d. (a) In-distribution: Evaluation of the datasets in which the models are trained. (b) Out-of-distribution: Evaluation of models, trained via random, in trajectories from other datasets.

### 5.3    HOW DOES NUNO ADDRESS THE MODEL EXPLOITATION PHENOMENON?

We assess how NUNO addresses the model exploitation phenomenon based on two aspects: (1) conservativeness of the reward function of pessimistic learned MDPs, and (2) prediction accuracy of learned dynamics models. Figures 4 and 5 evidence that NUNO enables less conservativeness and better accuracy over longer horizons. Figure 4 addresses the first aspect in two sets of D4RL tasks: random and medium-replay. Based on the gap between the groundtruth score and the pessimistic score, we observe that NUNO and NUNO[R] construct pessimistic learned MDPs that are less conservative than their counterparts in MOPO and TATU+MOPO, which use Gaussian ensembles. The only exception is hopper-medium-replay, which may be why TATU+MOPO and MOPO perform slightly better, as reported in Table 1. Model accuracy results in Figure 5 show that neural SDEs are significantly more accurate than a Gaussian ensemble over longer horizons.

## 6    CONCLUSION

We develop a novel uncertainty-aware offline model-based RL algorithm, NUNO, that learns a single dynamics model, in contrast to probabilistic ensembles in most existing work, as a neural SDE and addresses model exploitation phenomenon by penalizing and adaptively truncating model rollouts based on its uncertainty estimates. NUNO achieves this by imposing minimal prior physics knowledge into the drift term of a neural SDE as inductive bias and learning distance-aware uncertainty estimates via its diffusion term, which matches the dynamics' underlying stochasticity around the training data regime while providing high but bounded estimates beyond it. Through our empirical evaluations of NUNO in these benchmarks, we demonstrate that NUNO outperforms state-of-the-art methods, particularly in low-quality datasets with low state-action space coverage or suboptimal demonstrations ('random-v2' in D4RL and 'low' in NeoRL) by up to 93%. In tasks involving higher quality datasets, NUNO matches or exceeds the state-of-the-art performances in some environments by up to 55%.

**Limitations and future work.** Although we can extend our formulation to address partially observed Markov decision processes, our experiments utilize full knowledge of the system state in MuJoCo environments from both benchmarks. In the future, we aim to extend our uncertainty-aware approach to address different settings, e.g., environments with image observations. Additionally, future work can investigate formally proving properties of our distance-aware uncertainty estimator and extend our formulation for non-Euclidean state-action spaces by adjusting the distance metrics accordingly.

## REPRODUCIBILITY STATEMENT

For the theoretical analysis of this work, we state all assumptions made in Section 4 and Appendix B. For all the hyperparameters and detailed settings of the experiments, please refer to Appendix C. Lastly, we put the core code of our approach in the supplementary details. The code includes dataloaders, execution code, and links to download all the datasets and models used.

### ACKNOWLEDGMENTS

This work is supported by the Air Force Office of Scientific Research (AFOSR) under the grant FA9550-19-1-0005 and the National Science Foundation (NSF) under grant number 2214939.

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

## A  RELATED WORK - EXTENSION

**Offline model-free RL:**  Model-free RL algorithms in the offline setting aim to learn an optimal policy within the available data coverage without learning a dynamics model. Existing methods focus on learning policies that stay close to the data logging policy and avoid out-of-distribution actions by constraining the policy explicitly to the data logging policy Wu et al. (2019); Fujimoto et al. (2019); Fujimoto & Gu (2021), importance sampling Precup et al. (2001); Sutton et al. (2016); Gelada & Bellemare (2019); Nachum et al. (2019a); Rashidinejad et al. (2023), learning conservative value functions Nachum et al. (2019b); Kumar et al. (2020); Kostrikov et al. (2021), and uncertainty quantification Kumar et al. (2019); Agarwal et al. (2020); An et al. (2021); Wu et al. (2021); Bai et al. (2022). Although they refrain from the computational expense of learning a dynamics model, model-free approaches commonly struggle when the data logging policy is sub-optimal because optimal actions become out-of-distribution.

## B  DISTANCE-AWARE UNCERTAINTY ESTIMATOR

### B.1  PROOF OF LEMMA 1

This section provides results supporting the claim of Lemma 1. To this end, we analyze the extrema of the optimization problem

$$\bar{\eta}_\phi = \operatorname{argmin}_\eta \mathbb{E}_{(\mathbf{s},\mathbf{a})\sim\mathcal{D}}\big[\mathbb{E}_{(\mathbf{s}',\mathbf{a}')\sim\mathrm{Uniform}(\mathcal{S}\times\mathcal{A})}[\eta(\mathbf{s}',\mathbf{a}') - \|(\mathbf{s},\mathbf{a}) - (\mathbf{s}',\mathbf{a}')\|]^2\big]. \tag{9}$$

Under the assumption that $\mathcal{S} \times \mathcal{A}$ is compact, we can reformulate the optimization problem according to Fubini's theorem as

$$\min_\eta \mathbb{E}_{(\mathbf{s}',\mathbf{a}')\sim\mathrm{Uniform}(\mathcal{S}\times\mathcal{A})}\big[\mathbb{E}_{(\mathbf{s},\mathbf{a})\sim\mathcal{D}}[\eta(\mathbf{s}',\mathbf{a}') - \|(\mathbf{s},\mathbf{a}) - (\mathbf{s}',\mathbf{a}')\|]^2\big]. \tag{10}$$

Let $z = (\mathbf{s}, \mathbf{a})$ and $z' = (\mathbf{s}', \mathbf{a}')$. We can rewrite the objective function as

$$J(\eta(z')) = \int_{\mathcal{S}\times\mathcal{A}} \frac{1}{|\mathcal{S}\times\mathcal{A}|} \mathbb{E}_{z\sim\mathcal{D}}[\eta(z') - \|z - z'\|]^2 \, dz. \tag{11}$$

The extrema of the objective function are solutions of

$$\frac{\partial J(\eta(z'))}{\partial \eta(z')} = 0 \tag{12}$$

$$\Rightarrow \frac{\partial}{\partial \eta(z')}\big(\mathbb{E}_{z\sim\mathcal{D}}[\eta(z') - \|z - z'\|]^2\big) = 0 \tag{13}$$

$$\Rightarrow \mathbb{E}_{z\sim\mathcal{D}}[\eta(z') - \|z - z'\|] = 0. \tag{14}$$

Thus, by expanding the expectation, we have

$$\mathbb{E}_{z\sim\mathcal{D}}[\eta(z')] - \mathbb{E}_{z\sim\mathcal{D}}[\|z - z'\|] = 0 \tag{15}$$

$$\Rightarrow \eta(z') = \mathbb{E}_{z\sim\mathcal{D}}[\|z - z'\|]. \tag{16}$$

We can then conclude that the optimal solution $\eta$ is a convex function since it is a linear combination of convex functions.

Additionally, we have through Jensen's inequality that

$$\eta(z') = \mathbb{E}_{z\sim\mathcal{D}}[\|z - z'\|] \geq \|\mathbb{E}_{z\sim\mathcal{D}}[z] - z'\| = \|z_0 - z'\| = 0, \tag{17}$$

where $z_0 = \mathbb{E}_{z\sim\mathcal{D}}[z]$ is the state-action centroid of the dataset. Thus, the first property of Lemma 1 is proven.

Finally, let's prove that the negative of the gradient points inside the convex hull. By linearity of the gradient, we have

$$-\nabla\eta(z') = \mathbb{E}_{z\sim\mathcal{D}}\big[\|-\frac{z'-z}{\|z'-z\|}\|\big]. \tag{18}$$

This implies that for any point $z'$ that lies outside of the training dataset, the negative of the gradient is a non-negative combination of vectors $-\frac{z'-z}{\|z'-z\|}$ that points inside the convex hull of the dataset. This concludes the proof of Lemma 1.

### B.2 ILLUSTRATION OF THE DISTANCE-AWARE TERM ON TOY 2-D EXAMPLES

**Distance-aware estimator as model epistemic uncertainty estimator.** Most offline model-based reinforcement learning approaches employ either Monte Carlo (MC) dropout or model ensemble for epistemic uncertainty estimation. Although such approaches have demonstrated incredible results, Liu et al. (2020); Van Amersfoort et al. (2020) show that MC Dropout or model ensembles are unaware of the distance between unseen samples and training datasets, even in simple toy examples. Besides, these uncertainty estimators are parametric models targeted for reconstruction or regression objectives solely based on in-distribution data rather than directly tasked for uncertainty estimation. Therefore, they might discard relevant information, such as the distances between different samples or distances to out-of-sample data. We also refer to Figure 1 from Zhang et al. (2023a), where the authors demonstrate how these methods could not yield accurate distance-based uncertainty estimates.

Instead, using a distance-aware uncertainty estimator preserves the data's mutual relations while providing the ability to detect out-of-sample data. Besides, we can theoretically relate the problem of estimating the cross entropy between learned model dynamics and ground truth dynamics to calculating data point distances to a k-nearest neighbor clustering of the training dataset. Such cross-entropy is crucial for enforcing pessimism when training offline RL policies and for providing tight performance bounds. One of our goals is to design a parametric distance-aware uncertainty estimator that can efficiently cluster the dataset without performing k-nearest neighbor clustering and that can directly be embedded into the neural SDE formulation.

**Our term $\eta_\phi$ accurately provides distance-based uncertainty estimates.** We seek to demonstrate that the loss functions in equation 3 and equation 4 are sound and, upon convergence, provide a distance-based uncertainty estimate term $\eta_\phi$ that can efficiently cluster the training dataset. To this end, we generate three datasets of a two-dimensional system as illustrated in Figure 6. Our approach can cluster the training dataset in all examples while providing a clear delimitation between in-sample and out-of-sample data points.

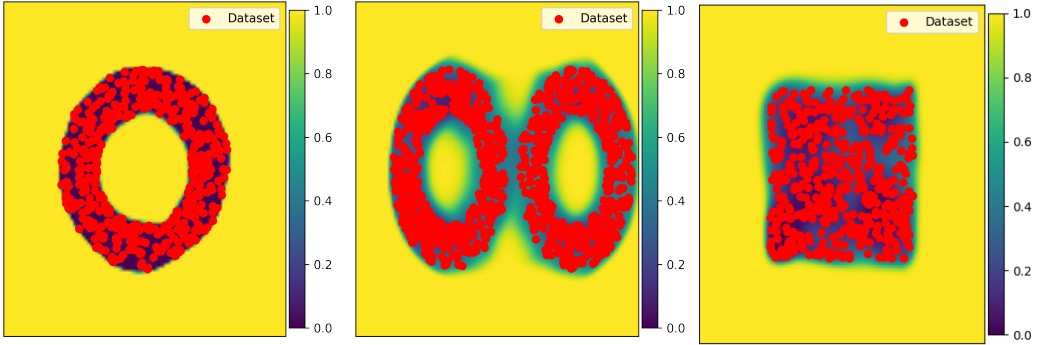

Figure 6: Visualization of the distance-aware uncertainty estimate $\eta_\phi$ on three generated datasets. The red points represent the state-action samples in the dataset. Yellow indicates high uncertainty, while dark blue represents low uncertainty. X and y-axis denote the states of the system, respectively.

## C EXPERIMENTAL DETAILS

### C.1 BENCHMARKS

We empirically evaluate `NUNO` in two continuous control benchmarks: D4RL and NeoRL. We utilize three MuJoCo environments from both benchmarks: halfcheetah, hopper, and walker2d. In D4RL, each environment comes with four types of datasets: (1) random-v2, where a randomly initialized policy collects the samples; (2) medium-v2, where an early-stopped policy trained via SAC Haarnoja et al. (2018) for one million steps is the data-logging policy; (3) medium-replay-v2, where the datasets comprises of the samples from the buffer of the early-stopped policy used for medium-v2; (4) medium-expert-v2, where half of the samples come from a medium-level policy and the other from an expert one. Our experiments use the v2 version of D4RL datasets. We report the results of MOBILE, RAMBO, and EDAC from their original papers, as the experiments were on v2 datasets.

For the rest, we provide the scores reported in TATU+MOPO paper (Zhang et al., 2023b), as our codebase is based on theirs. We note that MOBILE and EDAC train four independent runs for three million gradient steps, and RAMBO reports five runs of two million gradient steps, in contrast to five runs of one million steps for the rest.

In comparison, the NeoRL (Near real-world offline RL) benchmark consists of datasets collected by policies with validated performance. More specifically, NeoRL trains a policy via SAC until convergence and uses several checkpoints from the training to collect data. These checkpoints correspond to policies with three levels of sub-optimality: $25\%$, $50\%$, and $75\%$ of expert returns, which NeoRL calls low, medium, and high. The datasets we investigate consist of 1000 trajectories. We report the results of BC, CQL, and MOPO from the paper proposing NeoRL (Qin et al., 2022). For the rest, we provide the scores reported in the MOBILE paper (Sun et al., 2023). We exclude TATU+MOPO, TATU+MOPO, COMBO, and MOREL because NeoRL paper does not report any results for them, and also it would be extremely time-consuming to carry out a hyperparameter search for each approach. Similar to the experiments with D4RL, MOBILE results come from four independent training runs of three million gradient steps.

## C.2  POLICY OPTIMIZATION

Our implementation heavily relies on the codebase of Zhang et al. (2023b), which proposes uncertainty truncation, e.g., TATU+MOPO and TATU+COMBO. We use the default parameters of SAC described in Zhang et al. (2023b). We train RL agents on a cluster with NVIDIA RTX A5000 GPUs and an Intel(R) Xeon(R) Gold 6226R CPU @ 2.90GHz. Given a memory of approximately 6GB, a training run of 1 million gradient steps take around 12 hours.

## C.3  HYPERPARAMETERS OF NUNO

NUNO has four hyperparameters: real ratio $\beta$, rollout length $h$, CVaR coefficient $\alpha$ to set a truncation threshold, and uncertainty penalization threshold $\lambda_{\mathrm{pen}}$. The real ratio parameter $\beta$ refers to the ratio of samples from the real dataset in a mini-batch used to update the SAC policy. We set $\beta$ to 0.05, as TATU+MOPO, for all tasks in our experiments. For the rest of the parameters, we run a search over the following set of values: $h \in \{5, 10, 15, 20\}$, $\alpha \in \{0.9, 0.95, 0.98, 0.99, 1.0\}$, and $\lambda_{\mathrm{pen}} \in \{0.001, 0.1, 1\}$. Our hyperparameter search procedure starts by tuning for rollout length $h$ with $\alpha = 0.9$ and $\lambda_{\mathrm{pen}} = 0.001$. Using the best performing, namely, the highest human-normalized score yielding rollout length, we tune for $\alpha$. Finally, we run a search for $\lambda_{\mathrm{pen}}$. Table 3 reports the best-performing values for each task in our experiments. We use the same values for NUNO$^{\mathrm{R}}$.

## C.4  NEURAL SDE TRAINING

We implement all the numerical experiments using the python library JAX Bradbury et al. (2018), in order to take advantage of its automatic differentiation and just-in-time compilation features. We use Python 3.8.5 for the experiments and train all our models on a laptop computer with an Intel i9-9900 3.1 GHz CPU with 32 GB of RAM and a GeForce RTX 2060, TU106.

For training the neural SDE, we use randomly sampled sequences of horizon 2 for all the environments. We take the timestep of the ground truth environment and use it as the time step to integrate the neural SDE models. We use Euler-Maruyama as the integration scheme in all our experiments and generate one particle during each integration step to compute the expectation defined in $\mathcal{L}_{\mathrm{data}}$. For the regularization loss term $\mathcal{L}_{\mu}$, we define $\mu_{\phi}(\mathbf{s}_t, \mathbf{a}_t) = e^{NN_{\phi}(\mathbf{s}_t, \mathbf{a}_t)}$ ensuring that the output is positive, where $NN_{\phi}$ is a neural network parametrized by $\phi$.

For the neural SDE architecture, we parameterize $\eta_{\phi}$ as a neural network with two hidden layers of size 64 with swish activation functions. We parameterize the uncertainty term $\sigma_{\phi}$ as a neural network with two hidden layers of size 256 with tanh activation functions. The reward's drift term $f_{\theta}^{\mathrm{reward}}$ is parameterized as a neural network with three hidden layers of size 64 with swish activation functions while the other drift terms are parameterized with three hidden layers of size 256 and swish activation functions. Finally, the strong convexity neural network is parameterized with two hidden layers of size 32 with swish activation functions.

Table 3: Hyperparameters of NUNO in D4RL and NeoRL MuJoCo tasks

| Task | $h$ | $\alpha$ | $\lambda_{\mathrm{pen}}$ |
|------|-----|----------|--------------------------|
| hc-random | 20 | 1 | 0.001 |
| hp-random | 10 | 1 | 0.001 |
| wk-random | 10 | 0.99 | 0.001 |
| hc-medium | 5 | 0.99 | 1 |
| hp-medium | 10 | 0.99 | 1 |
| wk-medium | 10 | 0.98 | 1 |
| hc-medium-replay | 5 | 0.9 | 1 |
| hp-medium-replay | 10 | 0.99 | 1 |
| wk-medium-replay | 10 | 0.95 | 1 |
| hc-medium-expert | 10 | 0.95 | 1 |
| hp-medium-expert | 10 | 0.99 | 1 |
| wk-medium-expert | 10 | 0.98 | 1 |
| hc-low | 10 | 1 | 1 |
| hp-low | 10 | 0.99 | 0.001 |
| wk-low | 5 | 0.99 | 0.001 |
| hc-medium | 10 | 0.99 | 1 |
| hp-medium | 5 | 0.9 | 0.1 |
| wk-medium | 5 | 0.99 | 0.001 |
| hc-high | 5 | 0.99 | 1 |
| hp-high | 5 | 0.9 | 0.1 |
| wk-high | 5 | 0.99 | 1 |

The distance-aware diffusion term is trained with a ball radius of 0.1 in all environments and a strong convexity coefficient of 1. We use $\lambda_{\mathrm{grad}} = 10^{-4}$ and 20 samples to obtain the state-action needed to enforce the strong convexity constraint.

**Training optimizer hyperparameters.** We use the *Adam* optimizer (Kingma & Ba, 2014) for all optimization problems. We use the default hyperparameters for the optimizer, except for the learning rate, which we linearly decay from 0.01 to 0.001 over the first 5000 gradient steps. We use early stopping criteria for all our experiments. We use a batch size of 128 for the neural SDE training.

## D   ABLATION STUDY FOR UNCERTAINTY ESTIMATORS IN POLICY TRAINING

NUNO incorporates the distance-aware uncertainty estimate $\eta_\phi$ into the offline RL framework to enforce conservatism in the learned policy. Specifically, NUNO uses $\eta_\phi$ to penalize and truncate the transitions generated by the learned neural SDE model during the RL policy training. In an ablation study, we investigate whether the choice of the uncertainty estimator impacts the learned policy. We compare NUNO, which uses the distance-aware uncertainty estimate $\eta_\phi$, corresponding to epistemic uncertainty, against NUNO[al], which utilizes the aleatoric uncertainty estimate $\sigma_\phi$ (5). We evaluate NUNO and NUNO[al] in two types of datasets of the D4RL benchmark: random and medium-expert. Random datasets have low-quality trajectories, as the data-logging policies are sub-optimal. At the same time, random datasets have high coverage, as the trajectories showcase random behavior. In comparison, medium-expert datasets have high-quality trajectories yet low coverage as the data-logging policies are not random, and they act expert-like. D4RL benchmarks do not consist of very noisy datasets. Hence, we expect to have low aleatoric uncertainty in both datasets. However, data coverage determines epistemic uncertainty. We expect low epistemic uncertainty in random datasets and high in medium-expert ones.

Table 4 shows the best average human-normalized scores NUNO and NUNO[al] achieve, whereas Figure 7 demonstrates their performance progression. In random datasets, NUNO and NUNO[al] both achieve SOTA results, with NUNO performing better in halfcheetah and walker2d. In contrast, NUNO[al] performs significantly worse in medium-expert datasets, except in hopper, where NUNO and NUNO[al] achieve similar scores. These results align with our expectations based on the coverage properties of random and medium-expert datasets. A critical remark is that the reward penalty coefficient $\lambda_{\mathrm{pen}}$ is set to a low value, $\lambda_{\mathrm{pen}} = 0.001$, in random datasets. Hence, NUNO and NUNO[al] practically do not penalize the agent, except when the uncertainty is estimated to be very high. In comparison, in medium-expert datasets, the reward penalty coefficient is $\lambda_{\mathrm{pen}} = 1$, hence they frequently penalize

Table 4: Average human-normalized scores of NUNO and NUNO$^{al}$ in D4RL benchmarks. Due to limited space, we use abbreviations of task and dataset names: hc = halfcheetah, hp = hopper, wk = walker2d; r = random, me = medium-expert.We report the mean and standard deviation (following $\pm$) of best scores among independent runs. Bold scores indicate the best for each task.

| Task & Data | hc-r | hp-r | wk-r | hc-me | hp-me | wk-me |
|---|---|---|---|---|---|---|
| NUNO | $52.7_{\pm3.4}$ | $73.2_{\pm9.8}$ | $27.7_{\pm0.9}$ | $97.0_{\pm0.5}$ | $112.2_{\pm0.3}$ | $113.2_{\pm0.5}$ |
| NUNO$^{al}$ | $50.6_{\pm2.8}$ | $71.7_{\pm9.8}$ | $18.3_{\pm1.7}$ | $10.5_{\pm0.4}$ | $112.6_{\pm0.9}$ | $48.3_{\pm11.7}$ |

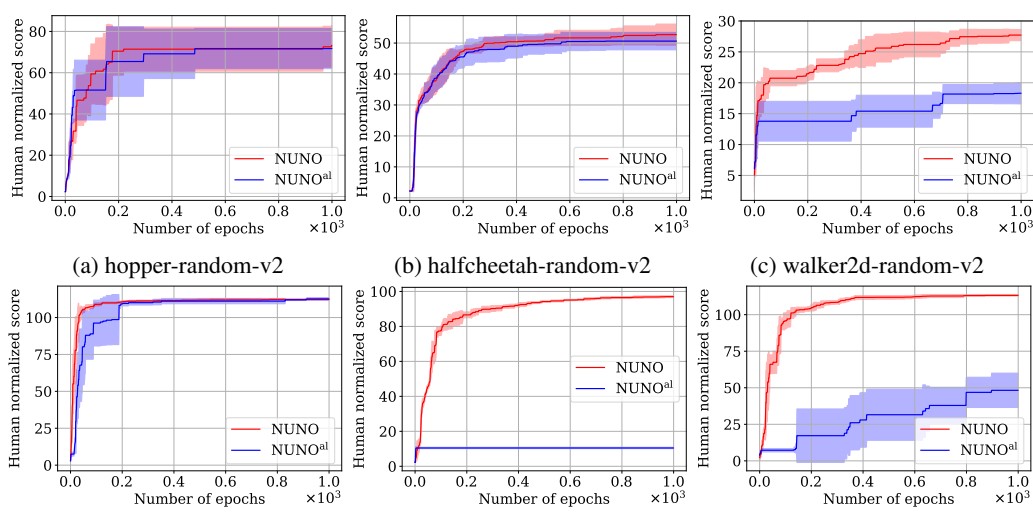

(a) hopper-random-v2     (b) halfcheetah-random-v2     (c) walker2d-random-v2

(d) hopper-medium-expert-v2     (e) halfcheetah-medium-expert-v2     (f) walker2d-medium-expert-v2

Figure 7: Ablation study: The impact of the choice of uncertainty estimator on policy learning.

the agent. As the epistemic uncertainty is expected to be high in this setting, the distance-aware uncertainty estimate is superior to the aleatoric uncertainty estimate.

# E DETAILED RESULTS

This section provides detailed results on model accuracy and training progression.

## E.1 MODEL ANALYSIS

Figure 8 provides model analysis results for D4RL Hopper and HalfCheetah (see Section 5 for results in D4RL Walker2d). In-distribution evaluation demonstrates how learned dynamics models perform over varying prediction horizons in trajectories from datasets with which the models are trained. In D4RL Hopper, probabilistic ensembles yield significantly higher prediction errors than neural SDEs as the horizon lengths increase. In D4RL HalfCheetah, the same pattern occurs, except in halfcheetah-medium-replay-v2, where all models provide low prediction error. Out-of-distribution evaluation assesses how learned dynamics models trained with low-quality datasets, i.e., random, perform in trajectories collected by behavior policies that are better than a random policy. All models perform well in trajectories from the random task, which is in-distribution. However, in D4RL Hopper, ensembles yield high prediction error in medium-replay and medium-expert as the horizon length increases. In D4RL HalfCheetah, we observe the same results, except in medium-replay.

## E.2 TRAINING PROGRESSION

Figures 9, 10, and 11 demonstrate the training progression in D4RL domains, Hopper, HalfCheetah and Walker2d, respectively. The first columns illustrate the progression of human normalized score in evaluation episodes ran after every epoch during training. In most tasks, NUNO and NUNO$^{R}$ achieve similar human normalized scores at the end of the training, with some exceptions such as hopper-

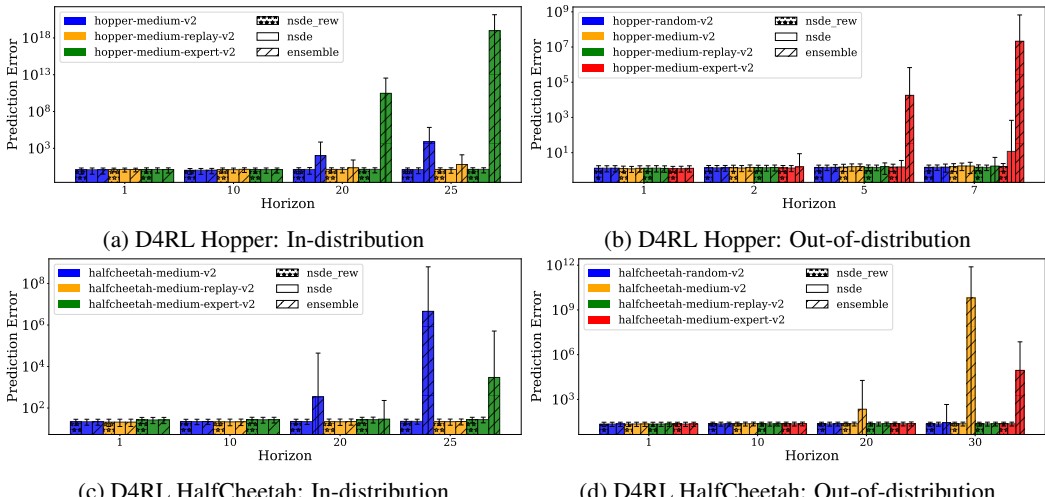

Figure 8: Model analysis: We illustrate the evolution of model prediction error in different datasets for D4RL Hopper and HalfCheetah. (a) In-distribution: Evaluation of the datasets in which the models are trained. (b) Out-of-distribution: Evaluation of models, trained via random, in trajectories from other datasets.

random-v2, halfcheetah-medium-v2, halfcheetah-medium-replay-v2, and walker2d-medium-v2. The second and third columns show the progression of the uncertainty estimates of neural SDEs trained in NUNO and NUNO$^R$, as well as those models' uncertainty estimates for random actions and actions from offline datasets. In random datasets, progress, random, and data curves are close to each other, as these datasets consist of trajectories from a random policy. In the rest of the tasks, neural SDEs can distinguish in-distribution actions (data) from out-of-distribution actions (random). Furthermore, as the trained policy progresses, the corresponding uncertainty estimates of neural SDEs approach the data curve. This is expected as neural SDEs generate synthetic trajectories close to offline data, and the policies' replay buffers are initially augmented with the offline dataset.

Figure 12 demonstrates the progression of human normalized score in NeoRL tasks. Like D4RL, in most tasks, NUNO and NUNO$^R$ reach similar scores. In Hopper-v3-Medium-1000 and Walker2d-v3-Medium-1000 NUNO outperforms NUNO$^R$. The opposite occurs in HalfCheetah-v3-Low-1000 and HalfCheetah-v3-Medium-1000.

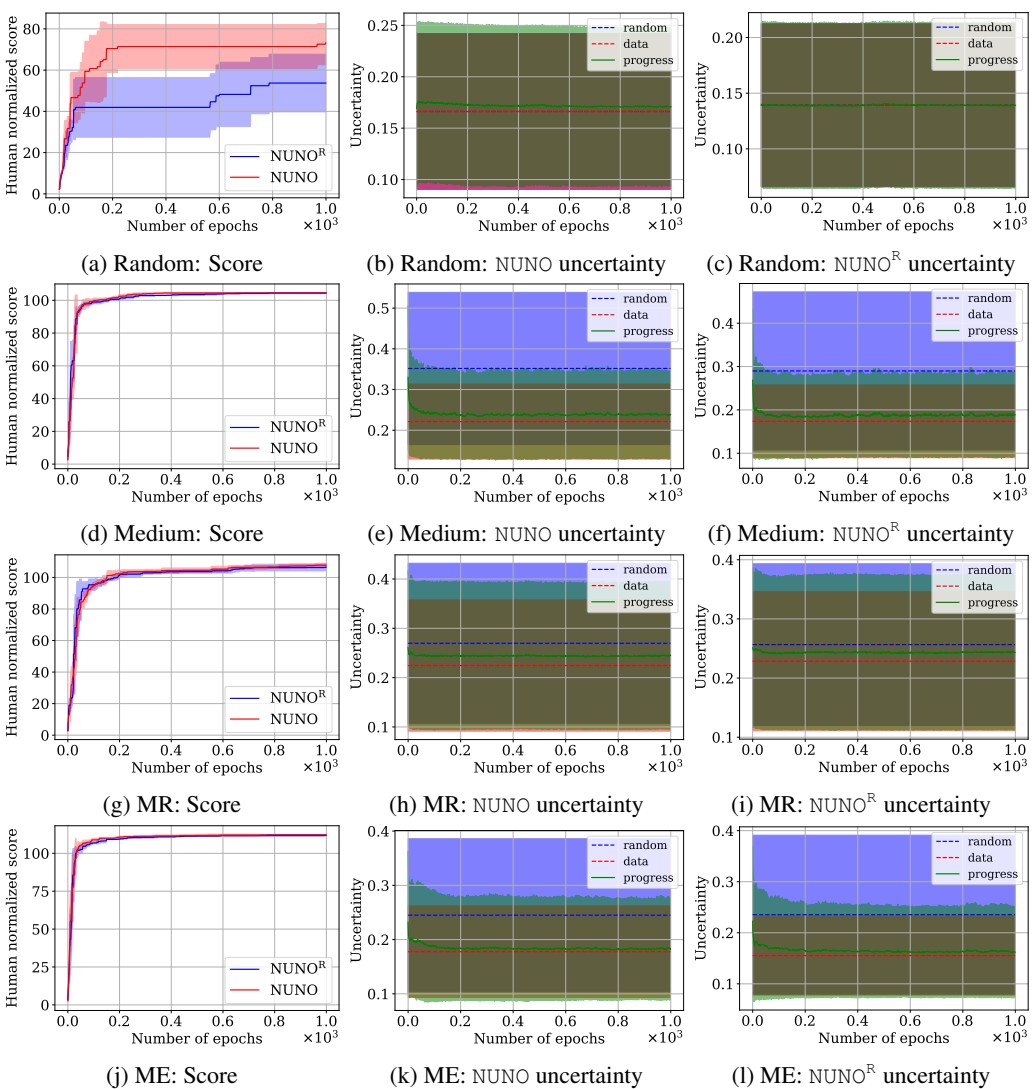

Figure 9: Training progression in D4RL Hopper: In the first column, we report the progression of human normalized score in evaluation episodes during training. In the second and third columns, we demonstrate how the uncertainty estimates of NSDEs in NUNO and NUNOR evolve when evaluated with trained policies' actions in one-step rollouts from states in the dataset. 'random' and 'data' refer to the uncertainty estimates of the learned model given actions from a random policy and the dataset, respectively. Each row corresponds to progression in a different task: random, medium, medium-replay, and medium-expert.

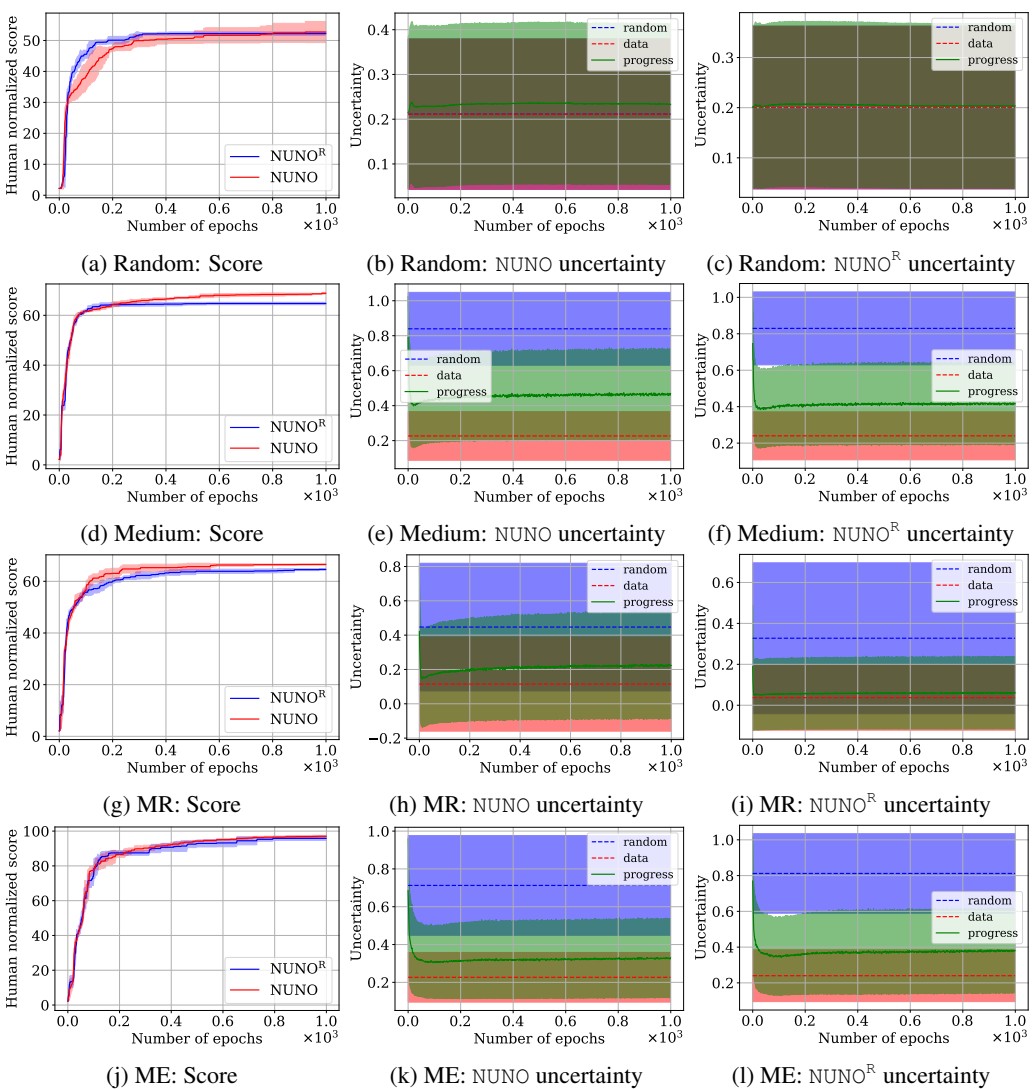

Figure 10: Training progression in D4RL HalfCheetah: In the first column, we report the progression of human normalized score in evaluation episodes during training. In the second and third columns, we demonstrate how the uncertainty estimates of NSDEs in NUNO and NUNO$^R$ evolve when evaluated with trained policies' actions in one-step rollouts from states in the dataset. 'random' and 'data' refer to the uncertainty estimates of the learned model given actions from a random policy and the dataset, respectively. Each row corresponds to progression in a different task: random, medium, medium-replay, and medium-expert.

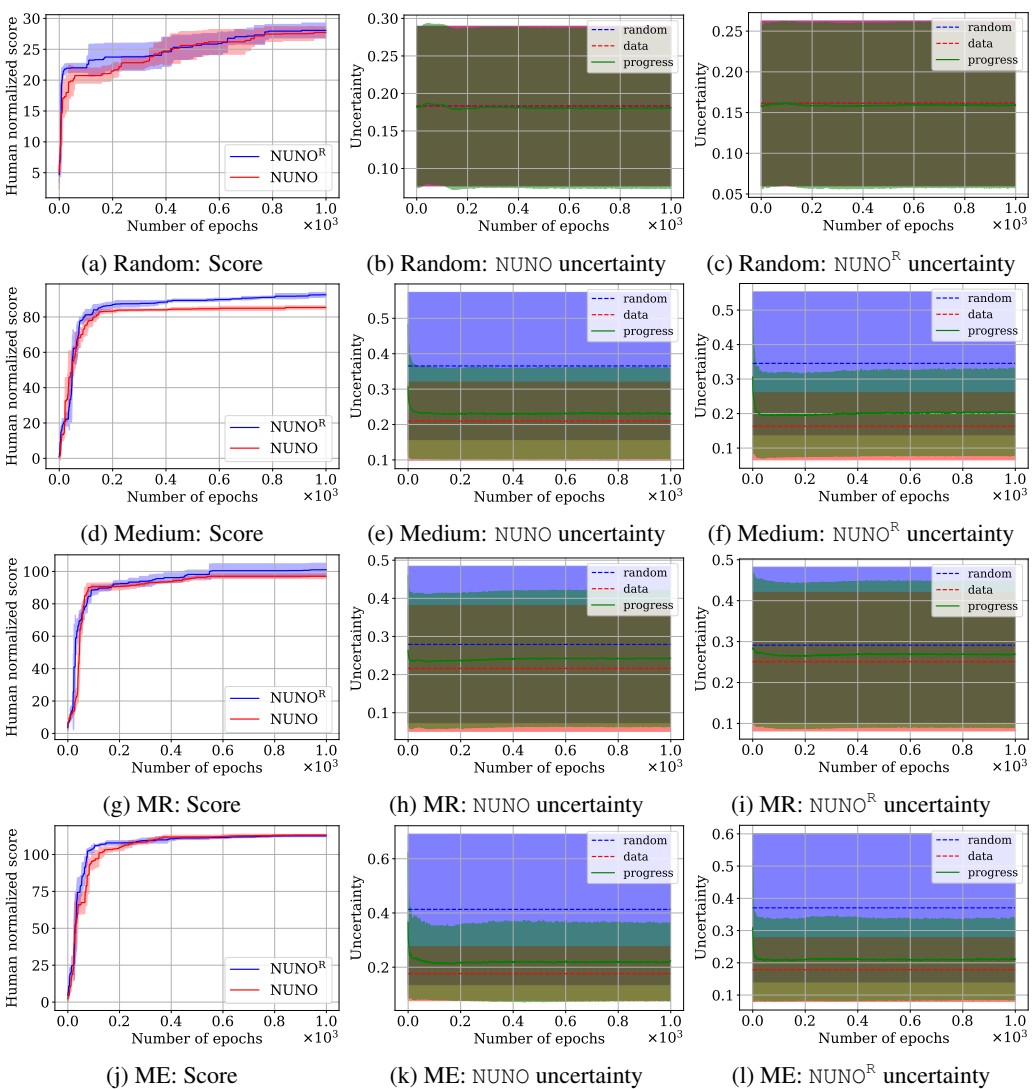

Figure 11: Training progression in D4RL Walker2d: In the first column, we report the progression of human normalized score in evaluation episodes during training. In the second and third columns, we demonstrate how the uncertainty estimates of NSDEs in NUNO and NUNO$^R$ evolve when evaluated with trained policies' actions in one-step rollouts from states in the dataset. 'random' and 'data' refer to the uncertainty estimates of the learned model given actions from a random policy and the dataset, respectively. Each row corresponds to progression in a different task: random, medium, medium-replay, and medium-expert.

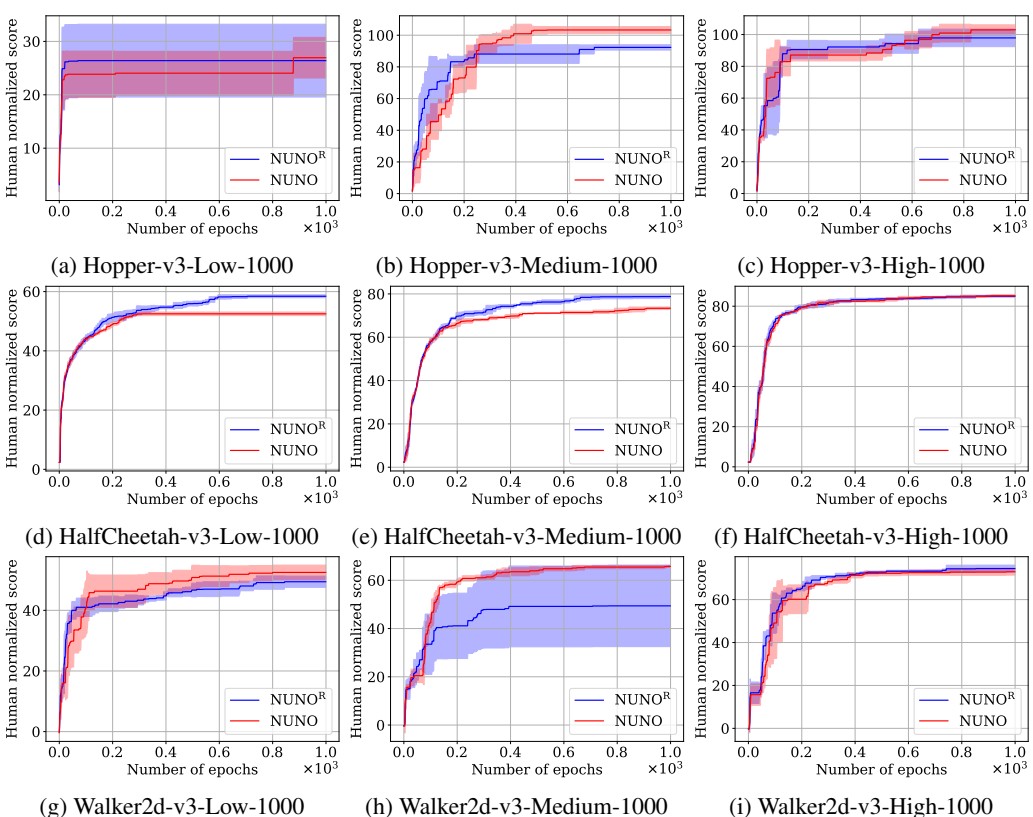

(a) Hopper-v3-Low-1000  (b) Hopper-v3-Medium-1000  (c) Hopper-v3-High-1000

(d) HalfCheetah-v3-Low-1000  (e) HalfCheetah-v3-Medium-1000  (f) HalfCheetah-v3-High-1000

(g) Walker2d-v3-Low-1000  (h) Walker2d-v3-Medium-1000  (i) Walker2d-v3-High-1000

Figure 12: Training progression in NeoRL tasks: Each subfigure reports the progression of human normalized score in evaluation episodes during training.

