# OpenReview forum: "Neural Stochastic Differential Equations for Uncertainty-Aware Offline RL"
_ICLR.cc/2025/Conference — ICLR 2025 Poster_

### Official Review · Reviewer_KRec · 2024-10-27

**Soundness:** 3
**Presentation:** 3
**Contribution:** 3
**Rating:** 6
**Confidence:** 4

**Summary:**

This paper introduces an offline model-based reinforcement learning method that can work well in low-quality data regimes while maintaining competitive performance on high-quality datasets. NUNO, the proposed  Neural Stochastic Differential Equations for
UNcertainty-aware Offline RL,  learns a dynamics model as neural stochastic differential equations (SDE), where the drift term integrates   physics knowledge to provide inductive bias, and the diffusion term estimates uncertainty to control the effects of uncertain predictions. Additionally, the paper addresses model exploitation by penalizing and truncating predictive rollouts, achieving strong results on several control benchmarks.

**Strengths:**

The paper proposes  a Neural Stochastic Differential Equation based method to address the important challenge of underrepresented state-action pairs in offline datasets and achieves strong experimental results. It has a solid theoretic foundation, and it is innovative and interesting.

The proposed method is well thought out, and the design for the uncertainty estimator has good local theoretical properties.

The paper is mostly well-written and easy to follow.

**Weaknesses:**

The proposed method relies on decomposing the state into position and velocity, and it imposes strong assumptions on the underlying environment dynamics (e.g., the dynamics must satisfy rigid body dynamics). I suggest that the authors also elaborate on how to apply the  physics-informed approach to other setting (not just position/velocity based).
While incorporating such prior physics knowledge is beneficial in some cases, clearly it is  not possible in all RL applications.

 Regarding the design of uncertainty estimator in 4.1, it would help to include simulation experiments under simplified settings of MDP and datasets to explicitly quantify the effects of the design choices.  This would shed light on the design of uncertainty estimator and hence the SDE model.

The computational overhead can be very high. Specifically, what is the computational overhead in training due to sampling around each data point for the uncertainty estimator? How does this affect the scalability of the method in high-dimensional datasets?

**Questions:**

In Equation 3, on the definition of \mathcal{L}_{\mu}, how is it guaranteed that the denominator $\mu$ is not zero in experiments?

In Equation 6, what is the intuition of the term log(det(\Sigma)) in the context of \mathcal{L}_{\text{data}}?

Could you provide examples of "unmodeled dynamics" as mentioned in Lines 342–343?

If the unmodeled dynamics are significant, would it still be beneficial to penalize the residual term?

---

> ### Author Response · Authors · 2024-11-24
> **Response to Reviewer KRec**
>
> We would like to thank the reviewer for their thought-provoking comments and questions that helped us improve our work in the rebuttal phase. Below, we first respond to their comments regarding the weaknesses and then their questions.
>
> **W1:** Relying of state decomposition and having strong assumptions on the environment.
>
> We emphasize that the position/velocity decomposition and rigid body dynamics are specific and minimal examples of prior knowledge that can be incorporated for the MuJoCo-based environments in the paper. These are typical environment choices for comparisons of offline model-based reinforcement algorithms. However, our physics-informed neural SDE model goes beyond rigid body dynamics and is even more relevant for various real-world applications, such as those involving aircraft, drones, or vehicles. In these scenarios, our framework can often leverage a range of a priori physical knowledge such as friction, gravity, and other constraints.
>
> Specifically, a priori physics knowledge can encompass (1) state parametrization, (2) structural knowledge of the vector field (right-hand side of the differential equations), and (3) a priori constraints that are satisfied by some terms in the structural knowledge of the vector field. In its most general form, structural knowledge defines the drift term of the neural SDE as $f_{\theta}(s, a) = F(s, a, g_{\theta_1}, ..., g_{\theta_N})$ where $F$ is a known function that composes a set of unknown terms or functions $g_{\theta_i}(s, a)$. This structural knowledge typically comes from applying first principles techniques, Kinematics, Kinetics, or rigid body dynamics. Furthermore, we may have known constraints that must be satisfied by $g_{\theta_i}$ in the form of $G(s, a,g_{\theta_1}, ..., g_{\theta_N}) \leq 0$, where $G$ is known. In this case, we can train the neural SDE to encode all the prior physics knowledge while minimizing constraint violations via an augmented Lagrangian method formulation.
>
> For instance, in the context of a vehicle operating at the limits of stability (racing or drifting), Kinematics and Kinetics can provide structural knowledge as $f_{\theta}(s, a) = M_{\phi}(s, a) F_{\theta}(s, a)$, where $M_{\phi}$ involves possibly unknown physical parameters like gravity, mass, inertia, and the vector $F_{\theta}$ represents the unknown and challenging-to-model forces resulting from the interaction between the ground and the vehicle. In addition, the friction law constrains even more $F_\theta$ through $F_x^2 + F_y^2 \leq \mu Fz$. This extra information is an example of physics knowledge we can apply to model learning.
>
> In environments where images define the observation space, we can encode the images into a latent space whose time evolution follows a neural SDE with prior physics knowledge, if available. However, we emphasize that most offline model-based RL approaches do not consider environments with images as observation space. Besides, it is not straightforward whether Gaussian ensembles are the right choice for modeling under such high-dimensional and non-continuous space.
>
> In environments without prior knowledge, the proposed neural SDE formulation is equivalent to a black box modeling under a time-varying assumption of the evolution of the observations.
>
> **W2:** Experiments in simplified settings to interpret the design choices for uncertainty estimators
>
> We have revised the paper and its appendix to provide additional details about the distance-aware uncertainty estimator, as well as extensive experiments to illustrate the properties of the estimator. Below is the summary of our changes:
>
> - We reference prior work in which distance-aware uncertainty estimators are theoretically and empirically demonstrated (on toy examples) to provide a better uncertainty estimate than model ensembling and Monte Carlo dropout, state-of-the-art models used in offline model-based reinforcement learning. **See Appendix B.2.**
> - We show on different dataset geometry of a 2-D system that our uncertainty estimator $\eta_{\phi}$ accurately clusters the training dataset and efficiently captures the notion of distance to the training dataset. These results provide further insights into the training mechanism of $\eta_{\phi}$ and how the design choices and desired properties encoded in (3) and (4) are empirically satisfied. **See Appendix B.2 and Figure 6.**
> - We provide new experiments in MuJoCo to demonstrate how the distance-aware uncertainty estimator of the learned models efficiently captures epistemic uncertainty and enforces pessimism during policy learning. In contrast, when using the aleatoric uncertainty term to enforce pessimism, we show that the resulting policy performs poorly in the ground truth environment. **See Appendix D and Figure 7.**

---

> ### Author Response · Authors · 2024-11-24
> **Response to Reviewer KRec - 2**
>
> **W3:** Computational overhead and scalability to high dimensional domains
>
> In all our experiments, sampling a single point around each training datapoint is enough for the distance-aware term $\eta_\phi$ to cluster the training dataset and capture distance-aware uncertainty accurately. However, we agree with the reviewer that the extra distance-aware training loss increases the computational complexity of our approach as the dimensionality increases. In the MuJoCo example in the paper, our method is computationally more efficient than the Gaussian ensemble as we train a single model to fit the data in contrast to an ensemble of $n$ models.
>
> **Q1:** How is it guaranteed for the denominator in Eq. 3 to be non-zero?
>
> In the experiments, we define $\mu_\phi$ as $\mu_{\phi}(s,a) = e^{NN_\phi(s,a)}$ ensuring that the output is always positve. Here, $NN_\phi$ is a neural network parametrized by $\phi$. We have clarified this design choice in the appendix of the revised paper.
>
> **Q2:** What is the intuition for $\log(\det(\Sigma))$ in the context of $\mathcal{L}_{\text{data}}$ in Eq. 6?
>
> The term $\log(\det(\Sigma))$ encapsulates the trade-off between accurately fitting the data and maintaining a model of appropriate complexity by acting as a regularizer that penalizes models with excessive variance. It helps prevent overfitting and promotes generalization, ensuring that the model remains both faithful to the data and as simple as possible.
>
> **Q3:** Could you provide examples of "unmodeled dynamics" as mentioned in Lines 342–343?
>
> For example, if the assumption of rigid body dynamics is incorrect, the residual term compensates for it. Besides, if the dynamics are not control-affine as assumed in (8), such residual term is required to learn the unmodeled part of the dynamics.
>
> **Q4:** If the unmodeled dynamics are significant, would it still be beneficial to penalize the residual term?
>
> Great question. If the unmodeled dynamics are significant, the training loss will indicate a failure to fit the data under such model assumption accurately. Then, one can tune the penalization terms to reduce the overreliance on the assumed model and improve the training loss. Such a hyperparameter tuning problem is also present when training an ensemble of Gaussian, and one needs to pick the number of models in the ensemble and the model architecture and size of each element in the ensemble to fit the data accurately.

---

> > ### Comment · Reviewer_KRec · 2024-11-27
> >
> > I'd like to thank the authors for their detailed response and great effort in addressing the comments. While some of the points have been clarified,  two main concerns raised in my review need to be addressed adequately.
> >
> > 1) From my earlier comments,  "The proposed method relies on decomposing the state into position and velocity, and it imposes strong assumptions on the underlying environment dynamics (e.g., the dynamics must satisfy rigid body dynamics). I suggest that the authors also elaborate on how to apply the physics-informed approach to other setting (not just position/velocity based). "
> > In the response and the revision the authors proposed to "train the neural SDE to encode all the prior physics knowledge while minimizing constraint violations via an augmented Lagrangian method formulation."  Given that one main idea of this paper is to leverage prior physics knowledge as inductive bias,  I suggest the authors provide more details how this training will be designed,
> > and also elaborate further on the limitation of this approach.
> >
> > 2)  In my earlier comments, I raised my concern on the computational complexity and asked "How does this affect the scalability of the method in high-dimensional datasets?" The response to this comment is elusive.

---

> > > ### Author Response · Authors · 2024-11-30
> > >
> > > Thank you for acknowledging our response and continuing our discussion. We want to respond to the reviewer's comments below.
> > >
> > > **Q1**: More details on training with physics constraints... and elaborate the limitations
> > >
> > > We thank the reviewer for the suggestion. It is important to note that the concept of incorporating prior physics and constraints into neural models for dynamical systems, specifically through the Augmented Lagrangian Method, is not new. We refer to a previous work [1] that provides extensive details on training deterministic neural ordinary differential equations (ODEs) with constraints. The work also demonstrates improved generalization and data efficiency and discusses the approach's limitations.
> > >
> > > The same technique can be straightforwardly extended to train neural SDEs with constraints, given an appropriate measure of constraint violation in a probabilistic setting. Due to space constraints in this paper and the additional focus on offline RL, we did not include these details in the main text. However, we have revised the Appendix to include a brief description of how general prior knowledge and constraints can be integrated into the training of neural SDEs while referring to the work above for further details.
> > >
> > > **Q2**: NUNO's scalability to high-dimensional systems
> > >
> > > We apologize if our previous response seemed rather unclear. We intended to convey that if there is a scalability issue as the dataset's dimensionality increases, then most existing methods (such as those based on Gaussian ensembles) would likely encounter similar challenges, given that these models are generally more computationally intensive than our approach. However, we have conducted additional experiments to provide further insights into our method's scalability.
> > >
> > > - **No Data Fitting**: We trained the distance-aware term only (as described in equations (3) and (4)). We measured the time it takes to perform one gradient step with a batch size of 128 across different environments, utilizing 20 samples around each state-action pair as defined in equation (4). We compared computation times for the Hopper (11 states and 3 actions), the HalfCheetah (17 states and 6 actions), and a randomly generated dataset with 100 states and 10 actions. For the Hopper, the gradient step took approximately 1.01 ms ± 0.1 ms; for the HalfCheetah, it was also about 1.02 ms ± 0.1 ms; and for the randomly generated dataset, it took around 4.2 ms ± 0.2 ms. This indicates that there is almost no increase in computation time when comparing the Hopper and HalfCheetah. At the same time, there is only a slight increase when moving to the randomly generated dataset. This suggests that adding the distance-aware loss term may not significantly impact the scalability of model training.
> > >
> > > - **Data Fitting**: We trained the full model (as shown in equation (7)) using the same setup and hyperparameters above, but now we focused on fitting the data and learning uncertainties. We also assessed how the integration horizon affects computation time. With a one-step integration horizon, the gradient step took 1.9 ms ± 0.2 ms on the Hopper and 2.2 ms ± 0.1 ms on the HalfCheetah. With a five-step integration horizon, the gradient step took 5.7 ms ± 0.2 ms on the Hopper and 6.2 ms ± 0.1 ms on the HalfCheetah. We observe that computation time for a fixed integration horizon increases with the number of states and actions; however, this increase is negligible, suggesting that our method scales well with the dimensionality of the dataset.
> > >
> > > **References**
> > > 1. Djeumou, F., Neary, C., Goubault E., Putot S., Topcu U. (2022). Neural Networks with Physics-Informed Architectures and Constraints for Dynamical Systems Modeling. Learning for Dynamics and Control Conference (L4DC).

---

### Official Review · Reviewer_jFi7 · 2024-11-03

**Soundness:** 2
**Presentation:** 3
**Contribution:** 2
**Rating:** 8
**Confidence:** 3

**Summary:**

In this work, the authors propose a method for offline model-based reinforcement learning that leverages uncertainty to enhance performance in settings with low-quality data or limited samples. The approach involves learning a model of the environment from samples using Neural Differential Equations (NDE), incorporating uncertainty and physics-informed learning. To account for uncertainty in the NDE, the authors introduce a parametric, distance-aware uncertainty estimator. Unlike kernel k-th nearest neighbor, this estimator avoids expensive search requirements while maintaining desirable properties such as monotonicity and boundedness. The estimator is used to limit the differential equation’s rollouts when distance is sufficiently high, restricting state-action paths to high-quality model rollouts and penalizing high-uncertainty paths, thereby encouraging the agent to favor paths where the NDE model is well-behaved. Simulations on D4RL and NeoRL, varying the quality of the policy used for offline training, demonstrate that the proposed method outperforms baselines in many cases. These baselines consist of other uncertainty-aware model-based methods that also penalize and truncate paths based on uncertainty.

**Strengths:**

- (S1): The problem of offline training with few samples and low-quality policies is a relevant one with significant applications in real-world settings.

- (S2): Extensive simulations, including hyperparameter searches and comparisons to baselines, demonstrate promising results.

- (S3): The manuscript is well-explained and relatively easy to follow, and the authors provide a thorough literature review that clarifies the position of their proposed method within the existing body of work.

- (S4): The proposed Distance-Aware uncertainty estimator is particularly interesting and may be useful for other problems. The authors ensure that their estimator is well-behaved through mathematical analysis, and they impose desirable properties on it through careful considerations in the optimization process.

**Weaknesses:**

- (W1): Some of the main contributions listed by the authors are not representative of the actual content presented in the paper. This could be improved by shifting the focus to other key aspects of the proposed method (see questions).

- (W2): The concept of uncertainty is underdeveloped in the manuscript, despite being a vital component of the proposed method. Although the authors clearly define distance-aware uncertainty, it is unclear what this concept represents in a broader context, given that the notion of uncertainty varies across applications and methods. This could be improved by examining the inner workings of their proposed uncertainty estimator in the simulated environments.

- (W3): While the results on the selected environments across different data regimes are promising (though other methods perform similarly to NUNO), there is a lack of exploration into the inner workings of the method. Obviously, simulations in these environments are computationally expensive, setting up a simpler problem with a known ground truth could enable multiple regime tests and ablation studies of the proposed method. This is especially important given the method's multiple components, and providing a thorough explanation of why and how the method works would significantly enhance the paper’s contribution.

**Questions:**

- Q1: How does incorporating prior physics knowledge differ from simply parameterizing the state space in a convenient way? Choosing a state representation is a standard design choice in most RL problems. How does this incorporation of prior knowledge apply to applications outside of MuJoCo? The parameterization in Equation (8) appears very specific to this case. Could you incorporate other types of physics knowledge, such as gravity, joint resistance, or environmental viscosity? One could argue that these environmental properties also represent physics knowledge.

- Q2: What are the implications of using dataset distance as an uncertainty estimator? The dynamical SDE described in Equation (5) can separate epistemic and aleatoric uncertainty in the dynamical model, where the epistemic uncertainty term depends on the distance-aware uncertainty estimation. It is possible that points further from the modes of the training set are influenced by noise, leading to high distance, and thus high epistemic uncertainty (since h in Equation 5 is monotonic). Conversely, a high-distance point could be "stable" (not noisy), with the dynamical model learning it well, so true epistemic uncertainty should be low for this point but is high according to the distance estimator.

- Q3: Related to the previous question, as the pessimistic reward is discounted by the distance-aware estimator, why isn’t aleatoric uncertainty considered here (as in Equation 5)? Highly noisy trajectories from the dynamical model cannot be learned by the agent, but they are not penalized, only those that the model hasn’t learned well enough based on distance are. This also relates to the previous question.

- Q4: How critical is this particular distance-aware uncertainty estimator? Why not use other uncertainty surrogates, such as prediction error from the dynamical model or the frequency of updates in a region (number of visits)? These other types of uncertainty estimation have been used in exploration-exploitation trade-offs in RL agents. This, and other ablation studies could be explored in a toy model.

- Q5: Typo in Equation 8: I believe the first variable should be s_{pos}? Additionally, the uncertainty penalization in Figure 1 doesn’t match the equation, and some notation in the figure is inconsistent with that used in the main text.

---

> ### Author Response · Authors · 2024-11-24
> **Response to Reviewer jfi7**
>
> We would like to thank the reviewer for their positive comments, valuable criticism, and thought-provoking questions that helped us improve the presentation of our work.
>
> First, we respond to the reviewer's comments and then their questions.
>
> **W1:** Some of the main contributions listed by the authors are not representative of the actual content(...)
>
> Could the reviewer clarify what is incomplete or misleading in the contributions we listed, please? We respond to their questions, as well, but are unsure which questions are related to this comment.
>
> **W2 and W3:** The concept of uncertainty is underdeveloped in the manuscript(...) and (...) there is a lack of exploration into the inner workings of the method.
>
> We thank the reviewer for raising these points. We have revised the paper and its appendix to provide additional details about the distance-aware uncertainty estimator and extensive experiments to illustrate the properties of the estimator. Below is the summary of our changes, which are provided in detail in the responses to Q1-Q4:
>
> - We reference prior work in which distance-aware uncertainty estimators are theoretically and empirically demonstrated (on toy examples) to provide a better uncertainty estimate than model ensembling and Monte Carlo dropout, state-of-the-art models used in offline model-based reinforcement learning. **See Appendix B.2.**
> - We show on different dataset geometry of a 2-D system that our uncertainty estimator $\eta_{\phi}$ accurately clusters the training dataset and efficiently captures the notion of distance to the training dataset. These results provide further insights into the training mechanism of $\eta_{\phi}$ and how the desired properties encoded in (3) and (4) are satisfied. **See Appendix B.2 and Figure 6.**
> - Extensive experiments in MuJoCo demonstrate how the distance-aware uncertainty estimator efficiently captures epistemic uncertainty. We also show that such an estimator enforces pessimism during policy learning in contrast to using aleatoric uncertainty, which we show cannot be used to enforce pessimism. **See Appendix D and Figure 7.**
>
> **Q1:** What are the differences between priori physics knowledge and conveniently parameterizing the state? What about applications other than MuJoCo? What about other types of physics knowledge?
>
> Incorporating a priori knowledge of physics goes beyond simply parametrizing the state. Section 4.2 presents a general formulation for integrating prior physics knowledge. We then emphasized that equation (8) reflects a specific and minimal understanding of rigid body dynamics, which we apply specifically for benchmark MuJoCo environments commonly used in offline model-based RL. However, our physics-informed neural SDE model is even more relevant for various real-world applications, such as those involving aircraft, drones, or vehicles. In these scenarios, our framework can often leverage a range of a priori physical knowledge such as friction, gravity, and other constraints.
>
> Specifically, a priori physics knowledge can encompass (1) state parametrization, (2) structural knowledge of the vector field (right-hand side of the differential equations), and (3) a priori constraints that are satisfied by some terms in the structural knowledge of the vector field. In its most general form, structural knowledge defines the drift term of the neural SDE as $f_{\theta}(s, a) = F(s, a, g_{\theta_1}, ..., g_{\theta_N})$ where $F$ is a known function that composes a set of unknown terms or functions $g_{\theta_i}(s, a)$. This structural knowledge typically comes from applying first principles techniques, Kinematics, Kinetics, or rigid body dynamics. Furthermore, we may have known constraints that must be satisfied by $g_{\theta_i}$ in the form of $G(s, a,g_{\theta_1}, ..., g_{\theta_N}) \leq 0$, where $G$ is known. In this case, we can train the neural SDE to encode all the prior physics knowledge while minimizing constraint violations via an augmented Lagrangian method formulation.
>
> For instance, in the context of a vehicle operating at the limits of stability (racing or drifting), Kinematics and Kinetics can provide structural knowledge as $f_{\theta}(s, a) = M_{\phi}(s, a) F_{\theta}(s, a)$, where $M_{\phi}$ involves possibly unknown physical parameters like gravity, mass, inertia, and the vector $F_{\theta}$ represents the unknown and challenging-to-model forces resulting from the interaction between the ground and the vehicle. In addition, the friction law constrains even more $F_\theta$ through $F_x^2 + F_y^2 \leq \mu Fz$. This extra information is an example of physics knowledge we can apply to model learning.
>
> We refer to the method outlined in [1] as an example of integrating general structural knowledge and constraints within the specific context of neural ordinary differential equations for modeling dynamical systems.

---

> ### Author Response · Authors · 2024-11-24
> **Response to Reviewer jFi7 - 2**
>
> **Q2:** What are the implications of using dataset distance as an uncertainty estimator?
>
> The reviewer raises a great question about high-distance points (or clusters) that may be either "noise" or actual "stable" data. In both cases, we argue that our distance-aware uncertainty term would either classify them as low-uncertainty areas or high-uncertainty areas in a manner consistent with state-of-the-art approaches based on model ensembling. Suppose a point is the result of noise and does not belong to any cluster in the training dataset. In that case, the point has a low density in the dataset, and model ensemble techniques will provide high epistemic uncertainty due to the disagreement in the predictions. Similarly, since our distance-aware estimator can efficiently cluster the training dataset (see **Appendix B.2 and Figure 6** for an illustration), those far-away points will have a high uncertainty estimate. Now, if the point is stable and not the result of noise, then by continuity, other "stable" points in the neighborhood of that point are in the training dataset and will form a new and separate cluster. Therefore, by the construction of the distance-aware uncertainty term, points in the cluster have low uncertainty estimates. If, for some unknown reason, we only have an isolated "stable" point in the training dataset, then due to that point's low mass in the training dataset, both the model ensemble and the distance-aware uncertainty estimators will provide high-epistemic uncertainty.
>
> We also want to emphasize that existing work [2-4] has shown that MC dropouts or model ensembles, popular modeling techniques in offline model-based RL, are unaware of the distance between unseen samples and training datasets, even in simple toy examples. Theoretical and empirical results have been provided to illustrate the benefits of using a distance-aware uncertainty estimate as an epistemic uncertainty estimator.
>
> **Q3:** Why isn't aleatoric uncertainty used in penalization?
>
> We consider this a design choice. Our intuition is not to penalize an agent with respect to aleatoric uncertainty because aleatoric uncertainty is a property of the dynamical system, not of data. When the offline dataset lacks coverage in certain areas, penalizing with respect to epistemic uncertainty is a way to signal the agent that the model may not have sufficient knowledge of said areas.
>
> As the reviewer suggests, there are many options for uncertainty estimators, especially in existing works that use Gaussian ensembles to learn dynamics models [5].
>
> **Q4:** How critical is this particular distance-aware uncertainty estimator? What about other uncertainty surrogates?
>
> In our global response, we provide an ablation study that evaluates the impact of the choice of uncertainty estimator to penalize/truncate rollouts during policy learning. More specifically, we look into two choices:
> 1) Distance-aware uncertainty estimate, which measures epistemic uncertainty, $\eta_{\phi}$, and
> 2) Aleatoric uncertainty $\sigma_{\phi}$.
>
> Our results showcase that in random datasets, the distance-aware uncertainty estimates $\eta_{\phi}$ performs similarly to aleatoric uncertainty, except in walker2d, where the epistemic uncertainty results in significantly better policies. However, in medium-expert datasets, the epistemic uncertainty estimator is clearly the best, as the aleatoric uncertainty estimator can result in trajectories with very low returns. One possible explanation behind these results is related to data coverage. Random datasets have better data coverage, as the data-logging policies behave randomly, resulting in diverse behaviors. In comparison, medium-expert datasets have better quality data yet low coverage hence the need for estimating epistemic uncertainty is more evident. For more details, please see the document shared in our global response.
>
> An uncertainty surrogate that estimates frequency of visitation may be unsuitable for continuous state space domains such as MuJoCo continuous control benchmarks. In addition, as [5] showcases, in offline model-based RL, surrogates based on learned dynamics models offer wide-ranging options. Our work follows a similar approach and learns uncertainty estimators as part of the dynamics model for more accurate next-state prediction. In addition, our work utilizes them to enforce conservatism in policy learning.

---

> ### Author Response · Authors · 2024-11-24
> **Response to Reviewer jFi7 - 3**
>
> **Q5:** Typos in Eq. (8) and Figure 1
>
> About Equation 8: The first variable is $s_{vel}$ because $f_{\theta}^{\text{pos}}$ computes the derivative of position. Notice that Eq. (8) is the derivative of the state.
>
> About Figure 1: We thank the reviewer for their attention to detail. Indeed, there are three typos in this figure: 1) $\textbf{s}'$ in uncertainty penalization should be $\textbf{s}$. 2) $\nu_{\phi}$ should be replaced with $\eta_{\phi}$, as in Eq. (8). 3) Epistemic uncertainty in uncertainty penalization should be  $\eta_{\phi}$. We made corrections in our new submission accordingly.
>
> - **References**
>     1. Djeumou, F., Neary, C., Goubault E., Putot S., Topcu U. (2022). Neural Networks with Physics-Informed Architectures and Constraints for Dynamical Systems Modeling. Learning for Dynamics and Control Conference (L4DC).
>     2. Liu, J., Lin, Z., Padhy, S., Tran, D., Bedrax Weiss, T., Lakshminarayanan, B. (2020). Simple and principled uncertainty estimation with deterministic deep learning via distance awareness. Advances in neural information processing systems (NeurIPS).
>     3. Van Amersfoort, J., Smith, L., Teh, Y. W., Gal, Y. (2020). Uncertainty estimation using a single deep deterministic neural network. International conference on machine learning (ICML).
>     4. Zhang, H., Shao, J., He, S., Jiang, Y., Ji, X. (2023). DARL: distance-aware uncertainty estimation for offline reinforcement learning. Proceedings of the AAAI Conference on Artificial Intelligence.
>     5. Lu, C., Ball, P. J., Parker-Holder, J., Osborne, M. A., & Roberts, S. J. (2021). Revisiting design choices in offline model-based reinforcement learning. arXiv preprint arXiv:2110.04135.

---

> ### Comment · Reviewer_jFi7 · 2024-11-27
>
> Thank you to the authors for their thorough answers to my concerns. The authors’ comments helped clarify some details and improve my understanding of the narrative of the paper. Overall, I think the paper is strong. It presents a well-studied method, and the authors have effectively addressed key details and the inner workings of their approach. This work opens many interesting avenues, which is a positive feature rather than a burden for the authors to develop every possible aspect of their method. That said, I believe there are two important points to clarify:
>
> > (W1): Contributions and Narrative Alignment
>
> Apologies, I realized I was not clear in my previous comment. When I stated that “the contributions listed by the authors are not representative of the actual content presented in the paper,” my intention was to highlight a different focus. I believe this paper makes a valuable contribution; however, I disagree with the authors on what that contribution is.
>
> In the contribution list, point (2) emphasizes the physics-governed dynamics while relegating the distance-aware uncertainty estimator to a secondary role within point (1). I would argue the opposite. The distance-aware uncertainty estimator is the standout idea in the revised manuscript, as it could potentially be applied to other problems. Meanwhile, the physics-aware feature seems more specifically engineered for the rigid body dynamics case, although it remains a meaningful contribution.
>
> Evidence for this change of focus includes the theoretical guarantees, the central role of the uncertainty measure in the model, and the additional results and descriptions provided for the estimator. In contrast, the functional form of prior physics knowledge is less explored and appears tailored to this specific scenario. I understand that the prior physics component of the proposed method can be adapted to other cases. However, the level of detail and relevance of this aspect seems less significant compared to the distance-aware uncertainty estimator. I would like to know your thoughts on this, considering the development of the distance-aware uncertainty estimator in the paper and the incorporation of prior physics knowledge.
>
> > Epistemic and Aleatoric Uncertainty
>
> Thanks to the author's responses, I now understand what initially confused me. However, I believe the use of uncertainty terms in the text requires greater care. I hope my argument here will be helpful.
>
> There are two types of uncertainty discussed in this work: epistemic uncertainty (which can be reduced through learning) and aleatoric uncertainty (intrinsic noise in the data that cannot be learned). In this work, they appear at two levels, each with its own epistemic and aleatoric components.
>
> - Distance-Aware Uncertainty Estimator Level:
> At this level, the estimator treats epistemic and aleatoric uncertainty from the data together as a single quantity. This is why I raised questions about aleatoric uncertainty (Q2 and Q3).
>
> - Model Level ($h_{\phi}$ and $\sigma_{\phi}$): Here, the stochastic ODE model separates epistemic (provided by the distance estimator) and aleatoric uncertainty. However, the text entangles these two pairs of uncertainties.
>
> The key distinction is that the agent is not directly exposed to the data uncertainty. Instead, the agent deals with model uncertainty, which separates epistemic and aleatoric components ($h_{\phi}$ and $\sigma_{\phi}$). In contrast, the distance-aware uncertainty estimator, which turns out to be the epistemic uncertainty of the stochastic ODE, does not have access to this separation and combines data epistemic and aleatoric uncertainty into one measure (as in the example of a point generated by noise, or being "stable"). While the separation of uncertainties in the distance estimator could be refined in future work, I think clarifying it in the current manuscript could help readers familiar with the literature on epistemic and aleatoric uncertainty.

---

> > ### Author Response · Authors · 2024-11-30
> >
> > We thank the reviewer for their response, helpful suggestions, and engaging with us for a detailed discussion. We want to respond to their discussion points.
> >
> > > (W1): Contributions and Narrative Alignment
> >
> > Thank you for the clarification. We agree that we should highlight our contribution via the distance-aware uncertainty estimator, and we understand why the list of contributions in the introduction is inadequate to do so. For that reason, we made the following changes:
> >
> > 1) In the abstract, we added a sentence preceding the last sentence to highlight our theoretical results:
> >
> > - > Our theoretical results show that penalization via a distance-aware uncertainty estimator incentivizes the policy to stay close to the offline data.
> >
> > 2) In the introduction, we added another contribution to the list, preceding the one on imposing structure on the drift term:
> >
> > - > We theoretically show that penalization via a distance-aware uncertainty estimator enables conservatism by encouraging the policy to stay close to the convex hull of the offline data.
> >
> > > Epistemic and Aleatoric Uncertainty
> >
> > Could the reviewer please clarify what they refer to by data uncertainty? We guess it is epistemic uncertainty, but we want to be sure.
> >
> > Before the discussion, we would like to clarify a couple of points. The reviewer correctly states that one can reduce epistemic uncertainty, unlike aleatoric uncertainty inherent in the groundtruth dynamics. However, it is not through learning but by collecting more training samples  [1,2]. In our setting of interest, the offline dataset is fixed, hence it is not possible to reduce epistemic uncertainty. Nevertheless, it is possible to quantify it; hence, we developed a distance-aware uncertainty estimator as a part of a neural SDE.
> >
> > We agree that there may be edge cases where distance-aware uncertainty estimates do not expose the agent to true epistemic uncertainty. The distance-aware uncertainty estimates $\eta_{\phi}$ will be zero around clusters in the offline data. As a result, around clustered points, the 'model level' uncertainty will be reduced to the aleatoric uncertainty estimates $\sigma_{\phi}$. Suppose that a noisy point is isolated such that it does not belong to any cluster. Then, both $\sigma_{\phi}$ and $\eta_{\phi}$  will be non-zero. The model will predict (as expected) high uncertainty, although NUNO won't be able to disentangle epistemic and aleatoric uncertainty. Nevertheless, we argue that enforcing conservatism via high 'model level' uncertainty around such points can still be beneficial, as the main objective is to incentivize the agent to stay close to the regions where data uncertainty is low.
> >
> > We added a discussion on such edge cases to the appendix to clarify the limitations of the distance-aware uncertainty estimator.
> >
> > - **References**
> >     1. Hüllermeier, E., & Waegeman, W. (2021). Aleatoric and epistemic uncertainty in machine learning: An introduction to concepts and methods. Machine learning, 110(3), 457-506.
> >     2. Wimmer, L., Sale, Y., Hofman, P., Bischl, B., & Hüllermeier, E. (2023, July). Quantifying aleatoric and epistemic uncertainty in machine learning: Are conditional entropy and mutual information appropriate measures?. In Uncertainty in Artificial Intelligence (pp. 2282-2292). PMLR.

---

> > > ### Comment · Reviewer_jFi7 · 2024-12-03
> > >
> > > I wanted to thank the author for their thoughtful response.
> > >
> > > I believe we are on the same page, and the problem might lie in semantics. I agree with the author's arguments, given their further explanations. After reviewing the discussion, I decided to increase my score. While there are areas for improvement, the authors have enhanced their exposition of results and explanations, opened interesting avenues, and clearly stated the limitations of their work. That said, I will attempt to restate my comment once more, as I think it represents an important distinction, though it is not critical to my decision.
> > >
> > > The distance-aware uncertainty estimator cannot separate noise uncertainty from epistemic uncertainty. The dataset's distance could be high due to noise (data aleatoric uncertainty) or a stable point (data epistemic uncertainty). Both uncertainties are contained in the $h_{\phi}$ term, which represents the model's epistemic uncertainty. However, when the neural differential equation is trained, if I understand correctly, the noise from the data is captured by $\sigma_{\phi}$. Therefore, the agent using the trajectories perceives *epistemic and aleatoric uncertainty from the NDE model* via $h_{\phi}$ and $\sigma_{\phi}$, but not directly from the data points. This makes sense, as the model generates the trajectories. The neural differential equation learns the environment dynamics and deals with the data uncertainty directly, while the agent learns from the trajectories generated by the neural differential equation, dealing with the uncertainty from the ODE model (these are the two levels of uncertainty, and why my initial confusion).
> > >
> > > I hope this explanation resonates. If not, I still think the current explanation by the authors is adequate given the discussion. I would recommend elaborating a bit more on this point in future iterations.
> > >
> > > Great work!

---

> ### Author Response · Authors · 2024-12-04
>
> Thank you for increasing your score! We appreciate your thought-provoking questions and comments, which led to discussions that eventually improved our paper in several parts, from the contributions to the limitations.
>
> We agree with your comment that there are cases where the distance-aware uncertainty estimator $\eta_{\phi}$ may not separate aleatoric and epistemic uncertainty. Your explanation resonates with our perspective. Hence, our newly added discussion covers these cases.
>
> Indeed, you are also correct that $\sigma_{\phi}$ is trained to capture the aleatoric uncertainty. Both $\eta_{\phi}$ and $\sigma_{\phi}$ are part of the neural SDE; hence, the trajectories generated by the model allow the agent to perceive both uncertainty sources, namely epistemic and aleatoric, respectively. NUNO utilizes the distance-aware uncertainty estimator to further enforce conservatism in policy training via uncertainty penalization and truncation. As our ablation study evidences, using the distance-aware uncertainty estimator $\eta_{\phi}$ to impose pessimism is more effective than the aleatoric uncertainty estimator $\sigma_{\phi}$. This is likely because the distance-aware uncertainty estimator represents the model uncertainty more accurately than the aleatoric uncertainty estimator, corresponding to the stochasticity of dynamics or noise in measurements.
>
> Thank you again!

---

### Official Review · Reviewer_wnnp · 2024-11-03

**Soundness:** 3
**Presentation:** 3
**Contribution:** 3
**Rating:** 6
**Confidence:** 4

**Summary:**

The paper presents NUNO, a method for offline reinforcement learning that uses neural SDEs to model dynamics with uncertainty. The SDE structure consists of a drift term that incorporates prior physics knowledge (as an inductive bias) and a diffusion term that captures uncertainty. Further, the uncertainty term is designed to account for both aleatoric and epistemic uncertainties explicitly to obtain accurate uncertainty estimates. By incorporating physics-based inductive biases and adaptive rollout truncation, NUNO mitigates model exploitation, where policies may overfit unreliable parts of the model.

**Strengths:**

**Strengths**
I like the paper overall. Here are some general comments.
- Impressive Results on Noisy Datasets: The paper achieves particularly strong performance on challenging, low-quality datasets, highlighting the robustness of NUNO’s uncertainty modeling in noisy conditions.
- Clear Presentation: The paper is well-written, with a logical structure that effectively communicates complex methods and results, making it accessible even with its technical depth.
- Novel Modeling Approaches: The decomposition of uncertainty into aleatoric and epistemic components, and learning them separately, stands out as an interesting and valuable contribution. This approach advances the understanding of uncertainty in offline RL and reinforces the model’s reliability.

**Weaknesses:**

**Weaknesses**
- Lack of Ablations for Design Decisions: A detailed analysis of each design decision, particularly the separate uncertainty terms, , would clarify their individual contributions to the model’s performance.
- Limited to Low-Dimensional Environments: The experiments are confined to low-dimensional MuJoCo environments, without tests on high-dimensional settings. Such environments may reveal limitations of the approach, where ensemble-based uncertainty methods could potentially perform better.
- Limited Scope on Partially Observable/Image-Based Domains: Although the paper notes the absence of experiments in partially observable or image-based environments, this remains a notable gap. These domains may challenge NUNO’s performance, and ensemble-based approaches might excel here due to their structure. Expanding experiments to these settings could provide a more comprehensive evaluation.

**Questions:**

same as weaknesses

---

> ### Author Response · Authors · 2024-11-24
> **Response to Reviewer wnnp**
>
> We thank the reviewer for their positive comments and fair criticism, which guided us to run an ablation study on our design choices and visualize their impact on toy domains. Below, we respond to the reviewer's comments.
>
> **W1:** Lack of ablation for design decisions.
>
> In our global response, we provide an ablation study that evaluates the impact of the choice of uncertainty estimator on policy learning. More specifically, we look into two choices:
> 1) Distance-aware uncertainty estimate, which measures epistemic uncertainty, $\eta_{\phi}$, and
> 2) Aleatoric uncertainty $\sigma_{\phi}$.
>
> Our results showcase that in random datasets, the distance-aware uncertainty estimates $\eta_{\phi}$ perform similarly to aleatoric uncertainty, except in walker2d, where the epistemic uncertainty results in significantly better policies. However, in medium-expert datasets, the epistemic uncertainty estimator is the best, as the aleatoric uncertainty estimator can result in trajectories with very low returns. One possible explanation behind these results is related to data coverage. Random datasets have better data coverage, as the data-logging policies behave randomly, resulting in diverse behaviors. In comparison, medium-expert datasets have better quality data yet low coverage; hence, the need for estimating epistemic uncertainty is more evident. For more details, please see the document shared in our global response.
>
> In addition, our global response provides experiments on different dataset geometry of a 2-D system to demonstrate that our distance-aware uncertainty estimator $\eta_{\phi}$ accurately clusters the training dataset and efficiently captures the notion of distance to the training dataset.
>
> **W2:** Focus on low-dimensional domains
>
> HalfCheetah, Hopper, and Walker2d environments in MuJoCo domains are standard evaluation benchmarks for offline model-based RL [1-4]. Therefore, we focus on these domains. Nevertheless, we acknowledge that datasets in the D4RL benchmarks may not reflect real-world characteristics from conservative data-logging policies. To address this, we also evaluate NUNO in NeoRL datasets designed accordingly. Our results show that NUNO outperforms all other approaches being assessed in terms of average among all datasets in both benchmarks.
>
> **W3:** Limited scope on partial observability and image-based domains.
>
> As the reviewer indicates, our discussion of this work's limitations already mentions that we do not address partially observable domains or domains with image-based observations. Extending our work to partially observable domains that do not have image-based observations is relatively more straightforward. We plan to extend our uncertainty-aware approach to image-based observations by learning dynamics on the latent space via a neural SDE, which is likely more challenging.
>
> - **References**
>     1. Yu, T., Thomas, G., Yu, L., Ermon, S., Zou, J. Y., Levine, S., ... & Ma, T. (2020). Mopo: Model-based offline policy optimization. Advances in Neural Information Processing Systems, 33, 14129-14142.
>     2. Yang, Y., Jiang, J., Zhou, T., Ma, J., & Shi, Y. (2021). Pareto policy pool for model-based offline reinforcement learning. In International Conference on Learning Representations.
>     3. Zhang, H., Shao, J., He, S., Jiang, Y., Ji, X. (2023). DARL: distance-aware uncertainty estimation for offline reinforcement learning. Proceedings of the AAAI Conference on Artificial Intelligence.
>     4. Sun, Y., Zhang, J., Jia, C., Lin, H., Ye, J. &amp; Yu, Y.. (2023). Model-Bellman Inconsistency for Model-based Offline Reinforcement Learning. Proceedings of the 40th International Conference on Machine Learning. 202:33177-33194

---

### Author Response · Authors · 2024-11-24
**Global Response**

We appreciate all reviewers for their thought-provoking comments and helpful questions. Thanks to them, we provided additional details about the distance-aware uncertainty estimator and extensive experiments to illustrate the properties of the estimator. The following list captures the changes made:
- [We ran an ablation study in MuJoCo domains](https://drive.google.com/file/d/1EV0zmIV8M-_PwrRTtPG8JV8exE2pMc84/view?usp=sharing) to demonstrate how the distance-aware uncertainty estimator efficiently captures epistemic uncertainty in contrast to aleatoric uncertainty, which we show cannot be used to enforce pessimism. **See Appendix D and Figure 7.**
- [We demonstrated on different dataset geometry of a 2-D system](https://drive.google.com/file/d/1EV0zmIV8M-_PwrRTtPG8JV8exE2pMc84/view?usp=sharing) that our uncertainty estimator $\eta_{\phi}$ accurately clusters the training dataset and efficiently captures the notion of distance to the training dataset. **See Appendix B.2 and Figure 6.**
- [We highlighted prior work](https://drive.google.com/file/d/1EV0zmIV8M-_PwrRTtPG8JV8exE2pMc84/view?usp=sharing) in which distance-aware uncertainty estimators are theoretically and empirically demonstrated (on toy examples) to provide a better uncertainty estimate than model ensembling and Monte Carlo dropout, state-of-the-art models used in offline model-based reinforcement learning. **See Appendix B.2.**
- We made a minor correction in the illustrative figure for NUNO. **See Figure 1**.
- We clarified details on loss terms and how to incorporate prior physics knowledge to address the reviewers' questions. **See Section 4.2. and Appendix C.4**

---

### Author Response · Authors · 2024-12-02
**A kind reminder**

We hope our rebuttal, responses during the discussion period, and the changes made to the draft have addressed the questions and weaknesses in your reviews. We are happy to address further questions or concerns about our contributions on the last day of the discussion phase. If our clarifications and the newly added results meet your expectations, we kindly request you consider revising the score before the discussion period ends today.

---

### Meta-Review · Area_Chair_ARX3 · 2024-12-18

**Metareview:**

This paper proposes an offline model-based RL approach that is robust in low-quality data regimes. To achieve this, it learns the environment as a neural SDE method that incorporates prior physics knowledge, and introduces a distance-aware uncertainty estimator to mitigate the model exploitation problem in the offline model-based RL approaches.

Reviewers appreciate its novelty in using neural SDEs to model dynamics with uncertainty. The method shows theoretical and empirical evidence of robust performance in low-data quality regimes. It also matches or surpassing baselines in high-quality settings.

Weaknesses of this approach includes the requirement of prior physics knowledge to learn the model and its limitation to low-dimensional fully-observed RL problems. These are acknowledged in the manuscript and authors' rebuttal.

**Additional Comments On Reviewer Discussion:**

Reviewers have some shared questions about the design choices of the proposed method, and its limitation in high-dimensional / image based environments. The authors provided additional ablation in the revised version to explain its distance-aware uncertainty estimator. They also acknowledge its limitation in the problem scope that requires prior physics acknowledge and low-dimensions.

Their rebuttal also provided more clarification to the details and narrative of the paper requested by reviewer jFi7 who raised the rating afterwards.

Reviewer KRec has a remaining concern on the computational complexity in high-dimensional datasets. The authors' response is not sufficient to clarify it based on the existing experiments on Hopper and HalfCheetah. It would be useful for the authors to clarify this limitation in a revision.

---

### Decision · Program_Chairs · 2025-01-22

Accept (Poster)